# Vigilance state dissociation induced by 5-MeO-DMT in mice
Benjamin J. B. Bréant [1,2,3] ✉, José Prius Mengual[1,2,3], Alexander Andrews [1,2],
Anna Hoerder-Suabedissen [1,2,3], Jasmin Patel [1], David M. Bannerman [4], Trevor Sharp [5] &
Vladyslav V. Vyazovskiy [1,2,3] ✉

Psychedelics lead to profound changes in subjective experience and behaviour, which are typically conceptualised in psychological terms rather than corresponding to an altered brain state or a distinct state of vigilance. Here, we performed chronic electrophysiological recordings from the neocortex concomitant with pupillometry in freely moving adult male mice following an injection of a short-acting psychedelic 5-methoxy-N,N-dimethyltryptamine (5-MeO-DMT). We report an acute induction of a dissociated state, characterised by prominent slow oscillations in the cortex and marked pupil dilation in behaviourally awake, moving animals. REM sleep was initially markedly suppressed, but was overcompensated in the subsequent 48 hours, while administration of 5-MeO-DMT immediately after sleep deprivation attenuated the subsequent rebound of sleep slow-wave activity. We argue that the occurrence of a dissociated state combining features of waking and sleep may fundamentally underpin the known and hypothesised effects of psychedelics — from dream-like hallucinations to reopening of the critical period for plasticity.

States of arousal determine many aspects of behaviour and sensory function critical for survival. Typically, brain states are classified based on electroencephalographic (EEG) recordings, which allow their categorisation into wake, non-rapid eye movement (NREM) sleep, and rapid eye movement (REM) sleep[1]. Wake is defined by a fast activity of low amplitude on the EEG, and the active behaviour of the animal. During NREM sleep, the animal is typically resting[2,3] and EEG is dominated by slow waves and spindles[4], while during REM sleep, sometimes referred to as "paradoxical" sleep, wake-like cortical EEG activity is accompanied with behavioural unresponsiveness and often profound muscle atonia [1,5]. In addition to these three fundamental global states of arousal, there is a variety of mixed, hybrid or entirely distinct states where features of waking and sleep coexist. For example, a slowing down of the EEG has been observed in awake animals after sleep deprivation[6], near lesion sites in patients with stroke[7], during hibernation[8,9], anaesthesia[10–12], brain injury, syncope[13], coma[14], in mouse models with deficient neurotransmission[15], as well as immature states in neonate rodents or preterm babies[16]. Even deafferented cortical slabs, organotypic neuronal cultures or brain slices can exhibit slow waves[17], which led to the proposal that they signify a so called "default state", to which the brain always gravitates in the absence of stimulation or an excitatory input necessary to maintain an awake state[18].

The effects of psychoactive drug administration are often described as producing an altered psychological state, for example, associated with a diminished or absent subjective experience[19]. Perhaps the best-known example thereof is the so-called serotonin psychedelics, such as lysergic acid diethylamide (LSD), psilocybin, or 5-methoxy-N,N-dimethyltryptamine (5-MeO-DMT), and pharmacologically distinct compounds such as ketamine, which produce profound psychoactive effects in humans. These drugs are being investigated as potential treatments for mental health disorders and have been shown in clinical trials to have a strong, beneficial long-term effect in treatment-resistant depressive disorders and anxiety disorders[20–23]. One proposed neurobiological mechanism of the therapeutic action of classical psychedelics and ketamine is their effect on neural plasticity, including acute changes in markers of synaptic efficacy or "reopening" of the critical period for synaptic plasticity and learning, typical for early ontogeny[24–27]. However, the possibility that what matters in this regard is an altered state of vigilance or arousal, induced by these compounds and corresponding to a highly plastic state, has not been considered.

Our initial report in the neocortex[28] followed by studies employing hippocampal recordings in rats[29] suggest that slow oscillations during the awake state might account for the psychoactive effects of psychedelics, including their influence on emotional regulation[30,31]. However, to our knowledge, no systematic investigation of the properties of the altered brain

[1]Department of Physiology, Anatomy and Genetics, University of Oxford, Oxford, UK. [2]Sleep and Circadian Neuroscience Institute, University of Oxford, Oxford, UK. [3]The Kavli Institute for Nanoscience Discovery, University of Oxford, Oxford, UK. [4]Department of Experimental Psychology, University of Oxford, Oxford, UK. [5]Department of Pharmacology, University of Oxford, Oxford, UK. ✉e-mail: benjamin.breant@icm-institute.org; vladyslav.vyazovskiy@dpag.ox.ac.uk

and behavioural state induced by psychedelics in mice has been undertaken. The motivation for this study was to address the bidirectional relationship between psychedelics and sleep, since sleep is known to be essential for brain development, neural plasticity, and mental health[16,32,33]. We found that in mice, 5-MeO DMT acutely induced a dissociated vigilance state, in which active behaviour and pupil dilation were associated with the occurrence of slow waves in the neocortex. We argue that this state is compatible with both well-known and proposed subjective effects of psychedelics, such as dream-like mentation and altered perception, and that it could also underlie the synaptic remodelling thought to be essential for their therapeutic effects.

## Results

### 5-MeO-DMT acutely alters cortical EEG

Following the intraperitoneal (i.p.) injection of 5 mg/kg 5-MeO-DMT in freely behaving mice, we observed the emergence of clear-cut slow oscillations on the cortical EEG (Fig. 1a, b). While brain signals were characteristic of NREM sleep, no other sleep-related oscillations such as spindles or sharp wave ripples were observed, and the animals were unequivocally awake, as confirmed by direct observations and video recording (Supplementary Movie 1) (Fig. 1.b).

Plotting the EEG spectrograms following vehicle and 5-MeO-DMT revealed that marked effects of the compound on cortical EEG activity lasted less than an hour in all cases (representative example: Fig. 1.c). Specifically, we observed that while the animals were awake and moving after the injection of 5-MeO-DMT, theta-frequency activity was replaced by slow frequencies for approximately 45 min before returning to levels comparable to vehicle (Fig. 1c). Further quantitative analyses determined that in the frontal derivation, the injection of 5-MeO-DMT resulted in a significant increase in EEG slow wave activity (SWA, 0.5–4 Hz) and EEG spectral power in the 15–20 Hz frequency range (Fig. 1d). In the occipital derivation, there was a significant increase in SWA and a significant decrease in theta frequency power (Fig. 1e). No difference was observed in the spindle frequency range, consistent with the absence of visually detected spindle

events. As non-oscillatory scale-free activity is considered an additional reliable marker of brain state[34,35], we compared the spectral slope calculated for the 20–120 Hz frequency range between the conditions. We found that in both derivations, the slope after 5-MeO-DMT treatment was comparable to normal wakefulness (Supplementary Table 1). We also observed that EEG power in faster frequencies ( > 30 Hz) was comparable to the awake state, despite the concomitant occurrence of slow waves, further suggesting that the state induced by 5-MeO-DMT has features of both waking and NREM sleep (Supplementary Fig. 1.a-d).

The differences in wake EEG power between 5-MeO-DMT and vehicle were observed primarily within one hour after the injection (Supplementary Fig. 1e-f). To further interrogate the effects of the compound on cortical activity, we calculated the modulation index across frequencies. No clear-cut changes were found in the coupling of slow waves and high frequencies, although the modulation index for theta/gamma coupling in the occipital derivation was slightly decreased after 5-MeO-DMT (Supplementary Fig. 2), consistent with the strong attenuation of theta oscillations.

### 5-MeO-DMT changes sleep architecture

Both NREM and REM sleep were initially suppressed after 5-MeO DMT (Supplementary Fig. 3.a-c), and the animals were behaviourally awake during the first 1 h interval after injection. While no major changes were observed in the subsequent sleep architecture and sleep stability (Supplementary Fig. 3.d-j), 48 h monitoring in a subset of animals revealed an increase in cumulative time spent in REM sleep over the first 24 h, with a further modest build up over the following 24 h period (Supplementary Fig. 3.k-n), resulting in overcompensation of REM sleep initially lost after the drug administration.

### 5-MeO-DMT effects on waking behaviour

In all animals, behaviour after 5-MeO-DMT administration remained largely normal, with no evident locomotor deficits. Specifically, we never

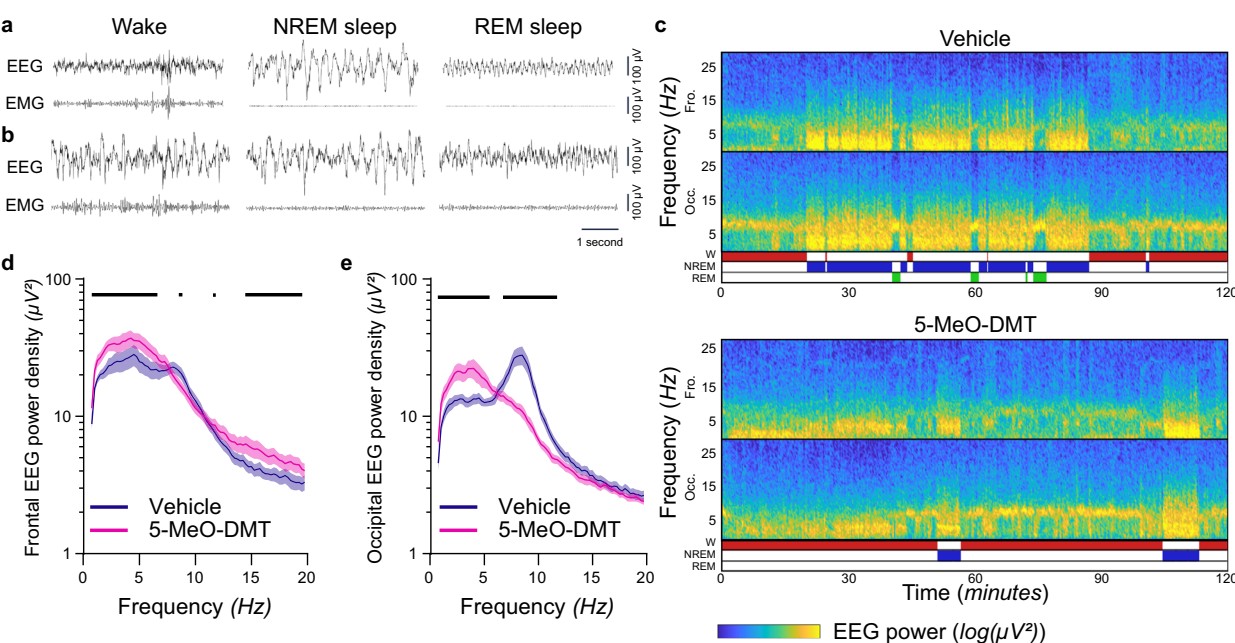

**Fig. 1 | Effects of 5-MeO-DMT on brain activity.** The injection of 5-MeO-DMT transiently increased EEG slow wave activity and suppressed theta activity during wakefulness. **a, b** Representative 5 s raw EEG and EMG signal during wake, NREM sleep and REM sleep during baseline (**a**) or following an injection of 5 mg/kg 5-MeO-DMT (**b**). Epochs occurring 5–60 min after the injection. **c** Representative spectrogram of the 120 min following an injection of vehicle or 5-MeO-DMT in the frontal and occipital EEG derivation. EEG spectral analysis of the episodes of wake occurring between 0–30 min after the injections shown for the frontal (**d**) and occipital (**e**) EEG signals. **d** $n = 8$; Mixed-Effect (ME) analysis, effect of frequency*condition $F_{152,1060} = 4.25$, $p < 0.0001$, Fisher's post hoc. **e** $n = 8$; ME analysis, effect of frequency*condition $F_{152,908} = 14.15$, $p < 0.0001$, Fisher's post hoc). Mean + SEM. A black horizontal line denotes a significant difference between vehicle and 5-MeO-DMT at the corresponding frequency (Fisher's LSD p < 0.05).

observed hyperactivity, flat body posture, catalepsy, hind limb abduction, or body tremors, and instead animals were engaged in exploratory behaviour, grooming, and nesting (Supplementary Movie 1, 01:31; 03:23). It is important to note that the injection of 5-MeO-DMT did induce head twitches, as characteristic of psychedelics (Supplementary Movie 1, 01:53) (Supplementary Fig. 4.a) and produced persistent exploration of the bedding immediately under the animal using their forelimbs (Supplementary Movie 1). To further characterise provoked behaviour and the motivational components of wakefulness, as well as to rule out behavioural abnormalities, we presented the animals with a plastic cup of sugar pellets and assessed their performance on a running wheel. 5-MeO-DMT increased the latency to cup interaction and pellet eating (Supplementary Fig. 4.b-c; Supplementary Movie 2), and shifted predominant behaviour to grooming or exploration. This might indicate a reduction in appetitively motivated behaviours, characteristic of the acute psychedelic state. We did not find any major change in the spontaneous wheel running behaviour that the control animals readily exhibited (Supplementary Fig. 4.d-g; Supplementary Movie 3), although, as the sample size in this experiment was low, no strong conclusions can be drawn at this stage.

### 5-MeO-DMT induces LFP slow waves accompanied with neuronal OFF-periods

As coexistence of active behaviours and global sleep-like brain signals is unusual, we next sought to rule out the possible occurrence of abnormal or artefactual brain signals, unrelated to physiological slow-wave activity[36]. To this end, we carefully inspected the local field potentials (LFP) and corresponding multi-unit activity (MUA) in the primary visual cortex (V1) recorded in a subset of animals. It is well established that slow waves during physiological sleep are accompanied by the occurrence of OFF-periods–generalised periods of synchronised neuronal silence, typically lasting 100-200 ms, when the recorded populations of neurons do not generate action potentials[37,38].

As expected, in the vehicle condition, LFPs and MUA showed typical signatures of both wakefulness and NREM sleep, with the latter characterised by frequent OFF-periods during LFP slow waves (Fig. 2.a-c, Supplementary Movie 4). 5-MeO-DMT injection also resulted in the occurrence of prominent LFP slow waves during wakefulness, which were invariably accompanied by neuronal OFF-periods, normally rare during wakefulness (Fig. 2d, e; Supplementary Movie 4). Plotting the distribution of slow waves as a function of their amplitude and duration revealed a marked redistribution towards higher amplitude values during wake after 5-MeO-DMT treatment as compared to the vehicle condition (Fig. 2.f), with no difference found in wave duration between NREM sleep, wake with vehicle and wake with 5-MeO-DMT (Fig. 2.g). However, as observed on the EEG, these slow waves coexisted with faster frequencies, consistent with the awake rather than the NREM sleep state.

### 5-MeO-DMT reduces the effects of sleep-deprivation on SWA rebound

Increased SWA both in waking and sleep is typical for elevated homeostatic sleep pressure[39]. Therefore, next we addressed the effects of sleep deprivation on 5-MeO-DMT-generated slow waves. We hypothesised that if slow waves after 5-MeO-DMT were reflecting increased sleep drive, they would be enhanced when the animals are experiencing high sleep pressure. To this end, we kept a group of animals awake for four hours, followed by an injection of the same dose of 5-MeO DMT as used before[32,39–41].

As expected, we observed that NREM sleep SWA after sleep deprivation was significantly increased compared to baseline in the vehicle condition (Fig. 3.a). Interestingly, elevated sleep pressure did not result in a further increase of SWA in awake animals following 5-MeO-DMT (Fig. 3b, c), suggesting that slow-waves induced by the drug correspond to an altered brain state rather than reflecting increased homeostatic sleep drive. However, unlike injections at light onset, 5-MeO-DMT administration at ZT4, following undisturbed sleep, resulted in significantly lower NREM SWA (Fig. 3d). Moreover, when administered immediately after sleep

deprivation, 5-MeO-DMT markedly attenuated SWA rebound during subsequent NREM sleep (Fig. 3e), suggesting an effect on the expression of sleep homeostasis.

### 5-MeO-DMT induces pupil dilation

The occurrence of slow waves has been linked to decreased levels of attention and alertness[42,43]. To measure the effects of 5-MeO-DMT on global arousal, we developed a device we called an oculometer. This was based on a miniature camera previously used in birds[44], that was fitted to the head-stage in order to track the state of the pupil in freely behaving mice. Previous studies in mice showed that movement or exploration - behaviours typically associated with increased levels of arousal are consistently accompanied by pupil dilation[45–47]. In our study, unrestrained mice wearing the oculometer exhibited normal spontaneous behaviours, including exploration, grooming, nesting, and sleeping (Fig. 4a; Supplementary Movie 5), while the quality of video recordings from the eye was sufficient for reliable tracking of the pupil and subsequent image analysis with DeepLabCut (Fig. 4b; Supplementary Movie 6).

Firstly, we replicated the effects of the injection of 5-MeO-DMT in animals wearing the oculometer (Supplementary Fig. 5), although the suppressing effects of the drug on theta-activity outlasted the effects on SWA in this cohort (Fig. 4c-e). The injection of 5-MeO-DMT led to a marked increase ( + 75%) in pupil size lasting for 10–15 minutes (Fig. 4f; Supplementary Movie 7), which correlated with the increase in frontal SWA and decrease in occipital theta-power (Fig. 4g-i). The latter was unexpected, as pupil size and elevated levels of arousal during wake behaviours have traditionally been associated with elevated hippocampal and cortical theta activity[48,49].

### Intracortical infusion of 5-MeO-DMT had no prominent effect

To begin addressing the origin of the changes in cortical and autonomic arousal induced by systemic administration of 5-MeO-DMT, we next performed intracortical infusions of the compound. With this experiment, we tested the hypothesis that the changes in cortical neural activity were driven by local 5-HT-mediated signalling vs global changes in neuromodulatory tone. The animals were implanted with cannulae, and 500 nL of 5-MeO-DMT (4.3 mg/mL) was infused in the primary somatosensory cortex (Supplementary Fig. 6), as has been done in an earlier study with a dose of 5-HT$_{2A/2c}$ agonist 1-(2,5-dimethoxy-4-iodophenyl)-2-aminopropane (DOI) that induced head twitches[50]. Contrary to those earlier findings, we did not observe any marked behavioural or EEG alterations, although a minor increase in EEG SWA was observed after the infusion (Supplementary Fig. 6). Further studies are necessary to clarify whether the prominent effects of systemic 5-MeO-DMT on brain state and arousal require the involvement of global, brain-wide circuitry, or depend on direct actions in other cortical or subcortical areas beyond the primary somatosensory cortex.

### 5-HT$_{1A}$ receptor antagonist prevents only some effects of 5-MeO-DMT

Unlike many serotonergic psychedelics, which preferentially target 5-HT$_{2A}$ receptors, 5-MeO-DMT is a non-specific 5-HT receptor agonist with a 100-fold higher affinity for 5-HT$_{1A}$ versus 5-HT$_{2A}$ receptors[51,52]. To investigate the role of 5-HT$_{1A}$ receptor-mediated neurotransmission in the effects of 5-MeO DMT on brain activity, a separate group of animals was injected with 1 mg/kg i.p. of the 5-HT$_{1A}$ receptor antagonist WAY-100635 (WAY) 15 minutes before the injection of 5-MeO-DMT. We observed that the animals injected with 5-MeO-DMT + WAY showed less activity and were often found staying immobile in one spot, exhibiting strong head twitch behaviour (Supplementary Movie 6). Furthermore, we observed that wake EEG SWA was further increased in the 5-MeO-DMT + WAY condition compared to both 5-MeO-DMT alone and vehicle conditions in both frontal (Fig. 5a) and occipital derivations (Fig. 5b), and this increase lasted longer frontally than in the 5-MeO-DMT alone condition (Fig. 5c), but not in the occipital derivation (Fig. 5d). Interestingly, the occipital theta activity,

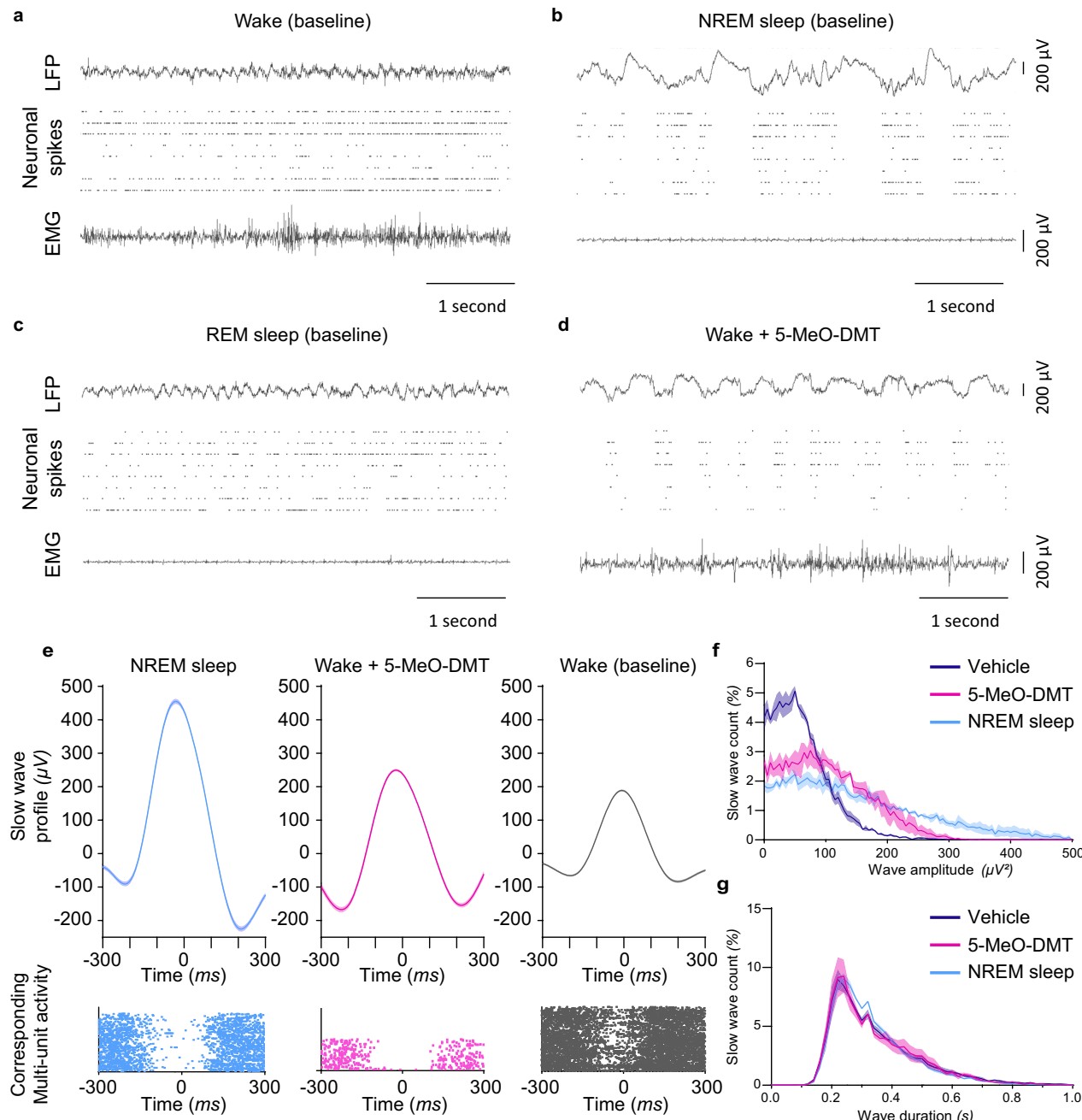

**Fig. 2 | LFP slow wave analysis following an injection of 5-MeO-DMT.** The injection of 5-MeO-DMT produced cortical LFP slow waves comparable to NREM sleep (**a**–**d**). Representative LFP traces with the corresponding neuronal spiking activity and EMG activity during baseline episodes of wake (**a**), NREM sleep (**b**) and REM sleep (**c**), or during wake in the 5–10 min following an injection of 5-MeO-DMT (**d**). **e** Average LFP slow wave (top) and corresponding multi-unit activity (bottom) in one representative animal during baseline NREM sleep, wake occurring 20 min after an injection of 5-MeO-DMT, and baseline wake (NREM sleep, $n = 272$;

wake + 5-MeO-DMT, $n = 505$; wake (baseline) $n = 119$). Distribution of the amplitude (**f**) and duration (**g**) of LFP slow waves during NREM sleep and during waking after the injection of vehicle or 5-MeO-DMT. **f** $n = 4$; NREM*5-MeO-DMT: $\chi^2_{55} = 8.50$, $p = 0.99$; vehicle*5-MeO-DMT: $\chi^2_{42} = 0.64$, $p < 0.001$; NREM*vehicle: $\chi^2_{42} = 92.11$, $p < 0.0001$; **g** $n = 4$; NREM*5-MeO-DMT: $\chi^2_{30} = 6.54$, $p > 0.99$; vehicle*5-MeO-DMT: $\chi^2_{30} = 1.80$, $p > 0.99$; NREM*vehicle: $\chi^2_{30} = 4.27$, $p > 0.99$). Mean + SEM.

which was strongly suppressed after 5-MeO-DMT alone, was no longer different from the vehicle condition after 5-MeO-DMT + WAY (Fig. 5e), as was also the case for pupil diameter, whose increase was abolished by the 5-HT$_{1A}$ antagonist (Fig. 5f). Thus, these data dissociate 5-MeO-DMT-induced changes in frontal SWA activity and pupil diameter, suggesting that only the latter effect is mediated by 5-HT$_{1A}$ receptors (Fig. 5g). These results indicate that different 5-HT receptor subtypes have unique and distinct contributions to different characteristics of the altered state of vigilance induced by 5-MeO DMT.

## Discussion

Here we report that 5-MeO-DMT induced a prominent change in EEG and LFP during wakefulness in mice, as reflected in the occurrence of slow waves associated with neuronal OFF-periods, along with marked suppression of theta-frequency activity. These observations support and extend the previous reports using the same psychedelic compound[28,29,53,54]. Interestingly, slow-wave activity was found to coexist with faster oscillations, typical of the awake state. In addition, the animals appeared behaviourally awake, manifesting normal behaviours, including grooming, exploring, and

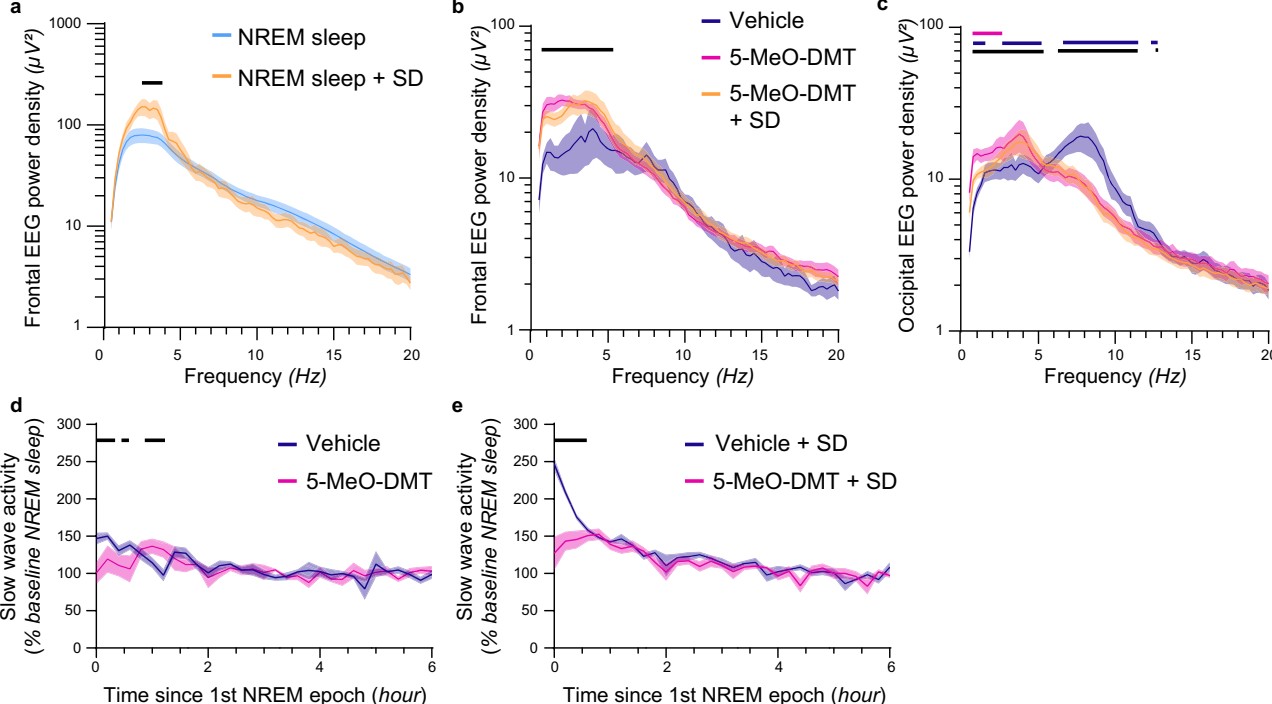

**Fig. 3 | Effects of sleep-wake history on the response to 5-MeO-DMT.** 5-MeO-DMT altered slow wave dynamics after sleep deprivation. **a** EEG spectral analysis during NREM sleep. A black horizontal line denotes a significant difference between NREM sleep and NREM + SD for the corresponding frequency ($n = 7$; ME analysis, frequency*condition: $F_{234,1476} = 6.82$, p < 0.0001; Fisher's LSD p < 0.05). Spectral analysis of the episodes of wake occurring between 0–30 min after the injections in the frontal (**b**) and occipital (**c**) derivations. A horizontal line denotes a difference between 5-MeO-DMT and vehicle (*black*), 5-MeO-DMT + SD and vehicle without SD (*blue*), and 5-MeO-DMT and 5-MeO-DMT + SD (*pink*) for the corresponding frequency (**b**. n = 8; ME analysis, frequency*condition: $F_{156,1013} = 7.40$, p < 0.0001; Fisher's LSD p < 0.05; **c** $n = 7$; ME analysis, effect of interaction $F_{156,936} = 10.39$, p < 0.0001; Fisher's LSD, p < 0.05). Frontal EEG SWA dynamics from the 1st NREM sleep episode onset following the injection of vehicle or 5-MeO-DMT at ZT = 4 without (**d**) or with sleep deprivation (**e**). A black horizontal line denotes a difference between 5-MeO-DMT and vehicle for the corresponding frequency ($n = 7$; ME analysis, frequency*condition: $F_{108,521} = 3.49$, p < 0.0001; Fisher's LSD p < 0.05). Mean + SEM.

running, despite sleep-like slow waves dominating the EEG and LFP signals. Most unexpected was the finding of a markedly increased pupil diameter immediately after drug administration, indicating increased levels of arousal. The main effects were short-lasting and dissipated within an hour, consistent with the kinetics of 5-MeO DMT[52,55–57]. However, delayed effects on REM sleep were observed. Furthermore, the 5-MeO-DMT-generated slow waves during wakefulness, although not impacted directly by sleep pressure, were followed by lowered SWA during NREM sleep if the drug was administered after sleep deprivation. Further investigation of potential mechanisms showed limited evidence for induction of slow waves by local application of 5-MeO DMT, and indicates that these slow waves, if anything, were enhanced in the absence of 5-HT$_{1A}$ receptor signalling. On the other hand, the suppression of theta activity and increased pupil dilation were effectively prevented by the injection of a 5-HT$_{1A}$ receptor antagonist.

The lack of any further increase of wake SWA when 5-MeO-DMT injection was given immediately after sleep deprivation may be interpreted as a ceiling effect. However, the finding that 5-MeO-DMT administered after sleep deprivation reduced the subsequent rebound of NREM sleep SWA suggests that 5-MeO-DMT-induced slow waves may contribute to the dissipation of homeostatic sleep pressure. We demonstrated that wake slow waves induced by 5-MeO-DMT are accompanied with population OFF periods, which were previously implicated in a wake-dependent increase in sleep drive[58]. However, whether the detection of OFF periods alone is sufficient to confirm that the underlying networks are in a "sleep-like" state can be debated[59], and whether this partial relief of sleep pressure also extends to the renormalisation of behavioural deficits remains to be determined. More generally, further experiments are necessary to fully characterise the slow waves, but also to address the possibility that high levels of wake SWA after 5-MeO-DMT provides an immediate and efficient recovery for brain networks similar to spontaneous sleep slow wave, or simply reflects a "default", idling cortical state where sleep pressure does not accumulate despite high levels of autonomic arousal and awake, active behaviour.

Our experiments using a 5-HT$_{1A}$ antagonist further highlight the complexity of the mechanisms behind the effects we report in this study. We predicted that co-treatment with WAY would counteract some of the effects of 5-MeO-DMT but did not expect that the main effect on cortical slow waves would not be among them. Visual observation of the animals following co-administration of 5-MeO-DMT and WAY suggested an exacerbation of some of the 5-HT$_{2A}$-mediated effects, such as the head twitch response[60,61], while wake SWA was also further enhanced. Since 5-HT$_{1A}$ receptors show reciprocal interaction with 5-HT$_{2A}$ receptors[62], we hypothesise that the occurrence of a dissociated state with high cortical SWA is mediated primarily by 5-HT$_{2A}$ neurotransmission. Further work is necessary to address the underlying circuit mechanisms, which could, for example, implicate cortical layer 5, known to play an important role in the generation of SWA[63], and shown to be essential in mediating the effects of certain psychedelics[64].

One interpretation of our data is that the changes in brain state and arousal we observe are regulated at the global rather than local cortical level. The regional differences in 5-MeO-DMT effects we report follow the regional distribution of 5-HT activity in a posterior-anterior gradient[65]. Thus, we believe that the brain-wide 5-HT circuitry is likely to be involved in the response to psychedelics administered systemically. Among the wide variety of subcortical neuromodulatory systems implicated in the regulation of global sleep-wake states, 5-HT has a special, albeit somewhat poorly defined role. It was thought to be wake-promoting, as neural activity of 5-HT neurons in the dorsal raphe nucleus (DRN) is elevated during wake and decreases during NREM sleep, with a complete cessation of activity

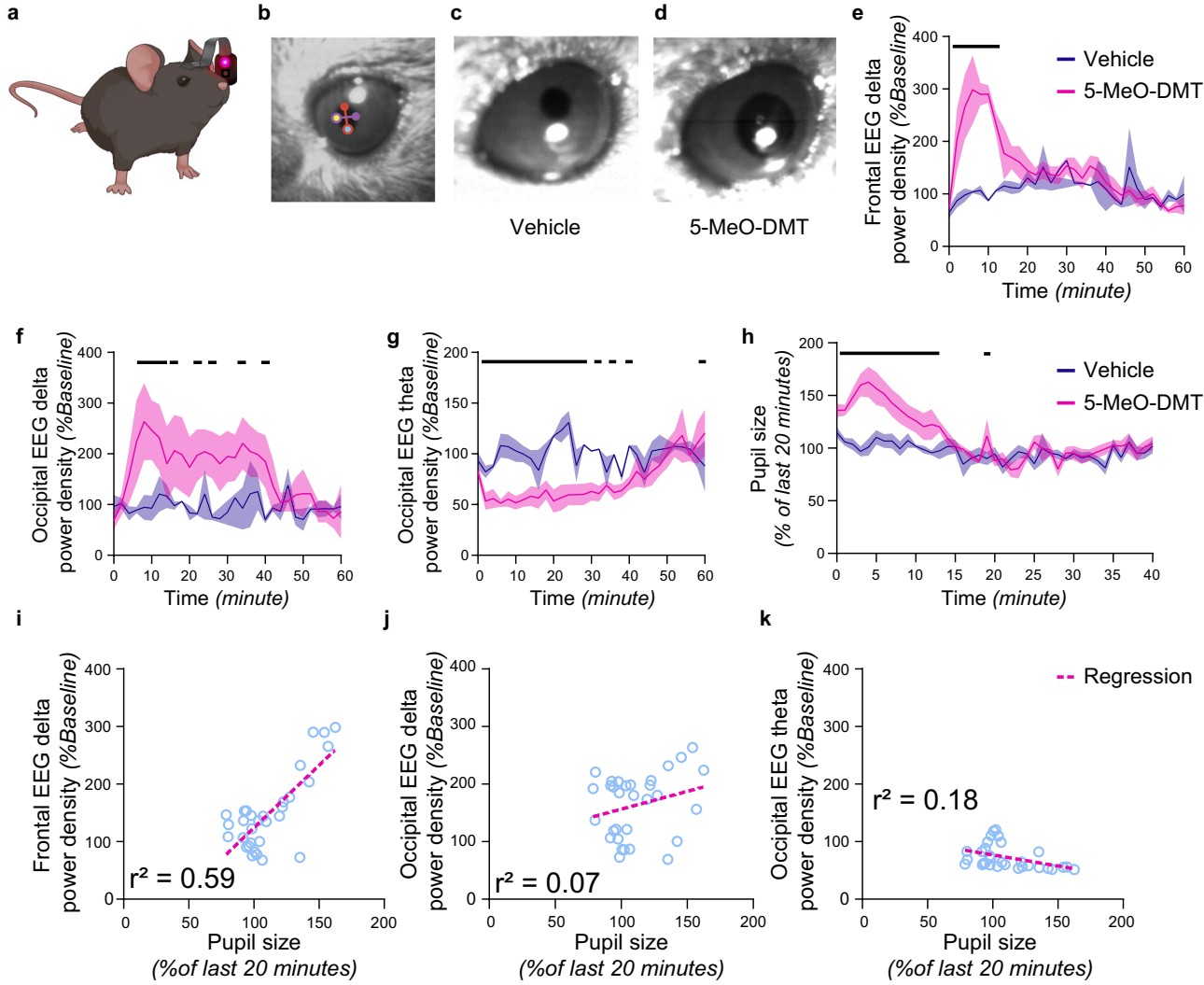

**Fig. 4 | Effects of 5-MeO-DMT on cortical activity dynamics and pupil diameter.**
5-MeO-DMT injections induce pupil mydriasis concomitant with EEG slow waves.
**a** Schematic of a mouse wearing the oculometer (Vanda Reiss). **b** Sample image from the oculometer software with the points manually scored at the north- (red), east- (purple), south- (cyan), and west- (yellow) most locations of the pupil. The North-South vector (red line) and East-West vector (purple line) used to calculate the pupil size are also represented. Representative image from the same animal taken 5 minutes after the injection of vehicle (**c**) or 5-MeO-DMT (**d**). Dynamics of frontal (**e**) and occipital (**f**) delta activity. **e** $n = 4$; ME analysis, effect of time

point*condition; $F_{30,7} = 5.19$, $p < 0.05$; Fisher's LSD $p < 0.05$; **f** $n = 4$; ME analysis, effect of time point*condition: $F_{30,15} = 2.98$, $p < 0.01$; Fisher's LSD $p < 0.05$).
**g** Dynamics of occipital theta activity ($n = 4$; ME analysis, effect of time point*-condition: $F_{30,6} = 5.20$, $p < 0.001$; Fisher's LSD $p < 0.05$). **h** Dynamics of pupil diameter ($n = 6$; ME analysis, effect of time point*condition: $F_{40,121} = 3.43$, $p < 0.0001$; Fisher's LSD $p < 0.05$). Correlation between pupil size and EEG delta (**i**–**j**) and theta (**k**) activity. (**i**. $n = 4$; Pearson's simple linear regression, $r^2 = 0.59$, $p < 0.0001$; **j** Pearson's simple linear regression, $r^2 = 0.18$, $p < 0.05$; **k** Pearson's simple linear regression, $r^2 = 0.07$, $p = 0.16$).

during REM sleep[66–68]. However, other studies suggest a role for 5-HT in hypnogenic processes[69,70], and evidence is emerging that DRN 5-HT neurons can be both sleep- and wake-promoting depending on their pattern of activity[71]. Thus, the 5-HT system is rather well-placed to regulate brain state quality or intensity rather than impose a unidirectional effect of all-or-none state switching, which is consistent with the occurrence of a mixed, dissociated state induced by psychedelics.

It is well recognised that wakefulness is a highly dynamic and heterogeneous state, sometimes sharing features with NREM sleep, for example, during quiet wakefulness or after sleep deprivation[72,73]. Our current observations highlight an underappreciated capacity of the brain to manifest hybrid states and support the call to reconsider the relationship between SWA, vigilance states, and sleep "depth"[6,15,43,74]. Here, we show that after 5-MeO-DMT treatment, slow waves can occur concomitantly with marked pupil mydriasis, typically associated with elevated autonomic arousal[45–47,75]. This suggests that local cortical mechanisms for generation and maintenance of slow-wave activity, sometimes considered a default

state of brain networks[15,17,76–79], are at least partially independent from the global control of a behavioural state. We argue that the state induced by 5-MeO-DMT may not only account for its well-described subjective effects in humans, such as altered perception and ego dissolution[80,81], but also contribute to the proposed long-lasting therapeutic effects of psychedelics[20,21,23,82,83]. Specifically, we propose that this property of brain networks might be essential for facilitating neural plasticity, as has been shown in early ontogeny, especially in relation to so-called critical periods of development[24,84], although any such connection remains to be investigated.

It is important to note that marked changes in the EEG, notably including an increase of slow wave activity, have been documented after other pharmacological compounds. Ketamine, a dissociative anaesthetic sharing some psychoactive properties with "classical" psychedelics[10,85–87], has been shown to increase slow wave activity in rats[88,89]. In awake cats and dogs, administration of the muscarinic receptor antagonists atropine and scopolamine induced a hybrid brain state, characterised by high-amplitude slow oscillations interspersed with wake-like gamma bursts[90–92]. Further work in humans has

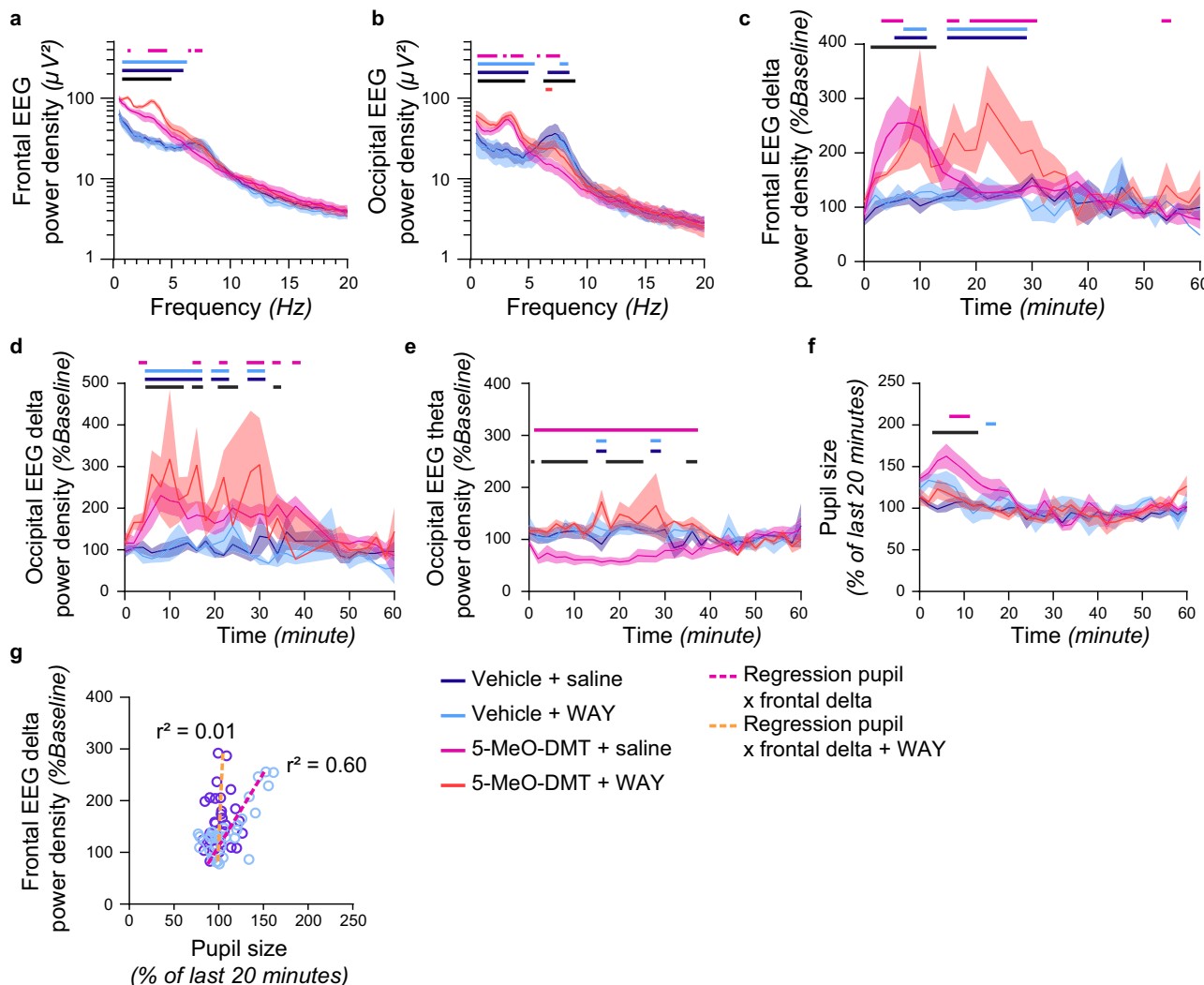

**Fig. 5 | The effects of 5-MeO-DMT are partially mediated by 5-HT$_{1A}$ receptors.** Spectral analysis of frontal (**a**) and occipital (**b**) EEG signals after an injection (**a**. $n = 6$; ME analysis, effect of frequency*condition: F$_{234,1009}$ = 11.49, $p < 0.0001$, Fisher's LSD $p < 0.05$; **b** $n = 5$; ME analysis, effect of frequency*condition: F$_{234,651}$ = 13.31, $p < 0.0001$, Fisher's LSD $p < 0.05$). Dynamics of EEG frontal (**c**) and occipital (**d**) delta (**c**. $n = 6$; ME analysis, effect of time point*condition: F$_{84,196}$ = 2.37, $p < 0.0001$; Fisher's LSD $p < 0.05$; **d** $n = 5$; ME analysis, effect of time point*condition: F$_{78,116}$ = 2.01, $p < 0.001$; Fisher's LSD $p < 0.05$). **e** Dynamics of occipital theta ($n = 5$; ME analysis, effect of time point*condition F$_{87,143}$ = 2.92,

$p < 0.0001$; Fisher's LSD $p > 0.05$). **f** Pupil size dynamics ($n = 5$; ME analysis, effect of time point*condition F$_{120,401}$ = 1.64, $p < 0.001$; Fisher's LSD $p > 0.05$). **g** Correlation between delta cortical activity and pupil dilation (5-MeO-DMT + saline; Pearson's simple linear regression, r$^2$ = 0.60, $p < 0.05$; 5-MeO-DMT + WAY; Pearson's simple linear regression, r$^2$ = 0.01, $p = 0.54$). Horizontal line denotes a significant difference between: (from bottom to top) vehicle and vehicle+WAY (*red*); 5-MeO-DMT and vehicle (*black*); 5-MeO-DMT + WAY and vehicle (*dark blue*); 5-MeO-DMT + WAY and vehicle+WAY (*light blue*); and 5-MeO-DMT and 5-MeO-DMT + WAY (*pink*). Mean + SEM.

examined antiepileptic drugs, GABA receptor agonists, and anticholinergic drugs, revealing a dissociated state arising from the paradoxical co-occurrence of slow waves with conscious wakefulness[92]. In summary, slow waves have been described outside the sleep context, and our results add serotonergic substances to an already diverse list. Notably, while atropine and scopolamine have been shown to induce spindles, this was not the case with 5-MeO-DMT, suggesting that distinct features of NREM sleep can be dissociated by pharmacological agents with different targets in the brain and possibly beyond.

The physiological meaning of the 5-MeO-DMT-generated slow waves remains to be determined. It has been proposed that sleep slow waves are functionally associated with structural synaptic changes, and more specifically, synaptic downscaling[93,94]. No consensus has been reached regarding the underlying cellular mechanism, but it is likely mediated, at least in part, by postsynaptic NMDA receptors and glycogen synthase kinase activity, coupled with rhythmic oscillations[94]. It remains to be determined whether 5-MeO-DMT-generated slow waves actively contribute to or reflect a state permissive of synaptic downscaling[95], which could result in a functional

decoupling and desynchronization between cortical regions[96]. If this were the case, it could provide one highly attractive and experimentally tractable mechanism for rapid antidepressant effects of psychedelics and other treatments associated with the induction of slow waves, such as ketamine[10]. Nevertheless, although hypothetical, we would not rule out the possibility that the state dominated by slow waves could provide a context for bidirectional synaptic plasticity, as observed during critical periods in early development when network slow waves are particularly prominent[16].

While it is tempting to describe the state induced by 5-MeO-DMT as "sleep-like"[28,29], it is important to acknowledge that it has many features not easily classifiable as any state of vigilance, and therefore, this terminology may be misleading. Specifically, fast frequencies characteristic of wake or REM sleep are prominent during the 5-MeO-DMT induced state, and slow waves induced by the compound could be partially dissociated from homeostatic sleep pressure. An alternative interpretation could be that slow waves induced by psychedelics reflect a brain state where coupling of the brain with interoceptive influences, such as respiration, is enhanced, as we

proposed earlier[30,31]. While the functional significance of this type of activity remains unclear, we speculate that it contributes to the context-dependent bidirectional network plasticity observed with other psychedelics[,23,27,94,97].

It is also important to stress that, in addition to increasing SWA, 5-MeO-DMT profoundly suppressed theta activity in the occipital derivation during active wakefulness, when theta is otherwise typically present in exploring animals. A similar effect has been reported in the hippocampus, even after controlling for locomotion[29]. Theta activity is thought to be generated by several subcortical and cortical circuits, including the hippocampal formation, where rhythmical activity has been linked to waking behaviour, notably pertaining to spatial and temporal processing[48,49,98,99]. Intriguingly, human psychedelic experience has been associated with altered processing of time and space[81], which may suggest that the suppression of theta activity observed here could be associated with a disruption of the processing of spatial and temporal information. This could, for example, reflect a dissociation between internally and externally generated information processing, leading to hallucinations. Although we did not observe abnormal behaviour or impaired locomotion in our implanted animals, video recordings were not of sufficient quality in all cases to permit detailed behavioural analysis. Therefore, at this stage, we cannot rule out reduced movement as a factor contributing to the suppression of the theta rhythm. However, as the occipital theta rhythm is restored after injection of WAY, and as the animals injected with 5-MeO-DMT + WAY remained largely immobile throughout, it is likely that the original suppression of theta is mediated through the activation of 5-HT$_{1A}$ receptors, independent of the general behavioural effects. As the specificity of WAY as a 5-HT$_{1A}$ receptor antagonist can be debated[100] other methods should be used to confirm these findings.

Finally, we observed a significant increase of REM sleep latency after 5-MeO-DMT, as was also reported after psilocin and Ayahuasca[101–103]. Whether this effect represents a direct pharmacological suppression of some of the essential defining characteristics of REM sleep, such as theta-activity, or reflects altered REM sleep regulation remains to be determined. One possibility is that the dissociated state induced by 5-MeO-DMT could replace some aspect(s) or functions(s) of REM sleep and thus reduce the homeostatic pressure to express REM sleep. REM sleep is thought to be primarily involved in emotional memory processing[104], which is likely relevant in psychedelic-assisted psychotherapy[105]. We speculate that de-contextualisation of emotional memories could be facilitated during induction of a sleep-like state, associated with a reopening of a critical period for neural plasticity[24,97]. These effects might be long lasting. While a rebound in REM sleep occurred within 6 h of the injections, 48 h monitoring shows a tendency for 5-MeO-DMT-injected mice to express progressively more REM sleep. This is a novel finding which is to our knowledge the first demonstration of delayed effects of psychedelics on sleep architecture.

In conclusion, we found that the injection of 5-MeO-DMT in mice induced a transient dissociated state of vigilance showing unequivocal features of wakefulness, such as active behaviour, high muscle tone, and dilated pupils, while brain activity showed slow oscillations typically associated with sleep. This paradoxical coexistence of wake and sleep characteristics mirrors that observed during paradoxical (REM) sleep and supports the notion that wake and sleep are not uniform, mutually exclusive phenomena. The effects of 5-MeO DMT we report here, such as the occurrence of slow waves in awake mice, the initial suppression of REM sleep followed by its delayed overcompensation, and the attenuation of sleep SWA rebound after sleep deprivation, may provide novel insights into the therapeutic properties of psychedelics. We speculate that the state emerging during 5-MeO-DMT administration has properties essential for heightened synaptic malleability and the reopening of the critical window for plasticity[106], which are posited to be essential for the therapeutic effects of psychedelics.

## Methods
### Animal husbandry
Adult male C57BL/6 J mice ($n = 42$, 9 to 15 weeks old) were kept singly housed in individual Plexiglas cages (20.3 cm × 32 cm × 35 cm) placed inside ventilated sound-attenuated Faraday chambers (Campden Instruments, Loughborough, UK), under a 12–12 h light-dark cycle (9 am–9 pm). The recording room was maintained at 22 ± 1 °C and 50 ± 20% humidity. Food and water were provided ad libitum throughout the experiment. A subset of animals was not implanted and were used to monitor head twitches ($n = 4$), to study the effects of injection on running wheel activity ($n = 4$), or to study feeding behaviour ($n = 4$). Sample sizes were determined based on previous publications[103]. All animals were monitored daily for unexpected weight loss or visible suffering as a marker of a humane endpoint. We have complied with all relevant ethical regulations for animal use. All procedures were performed under a UK Home Office Project License and conformed to the Animals (Scientific Procedures) Act 1986.

### Surgeries
Procedures were performed based on established protocols for device implantation in mice[103,107]. Prior to surgeries, mice ($n = 30$) were habituated to mash and jelly food and housed in individually ventilated cages. Surgeries were performed under isoflurane anaesthesia (4% induction, 1–2% maintenance). EEG screws were implanted above the right frontal cortex (2 mm anteroposterior, 2 mm mediolateral), right occipital cortex (anteroposterior 3.5 mm, mediolateral 2.5 mm), and left cerebellum for reference (Supplementary Fig. 7a).

In a subset of animals ($n = 10$), laminar probes (A1x16-3 mm-100-703-Z16, NeuroNexus) were implanted in addition to EEG electrodes as above, in the primary visual cortex (-3.4 mm anteroposterior, -2 mediolateral) and referenced to the cerebellum screw (Supplementary Fig. 7b).

In another subset of animals ($n = 4$), cannulae (C315G-005T GDE/ELE 26GA/.005" TUNG 8637 RECEPTACLE, Bilaney Consultants Ltd) affixed with two tungsten electrodes (4.8 mm cut below pedestal; 4.7 mm and 5 mm electrode Projection) were implanted unilaterally in the primary somatosensory cortex (-1.6 mm anteroposterior; +3.3 mm mediolateral; −0.5 mm dorsoventral from brain surface, with a 30° angle) (Supplementary Fig. 7.c). The location of the cannulae was then visually assessed by an experienced anatomist with cresyl violet coloration (Supplementary Fig. 8). EMG wires were inserted into the left and right nuchal muscles. Dental acrylic (Super Bond, Prestige Dental, Bradford, UK) was used to fix the implanted electrodes to the skull and to protect the exposed wires (Simplex Rapid, Kemdent, Swindon, UK).

A group of animals ($n = 12$) were also implanted frontally with a stainless-steel hexagonal nut (M2, DIN 934, RS Components) holder for the oculometer, a subset of which were also implanted with EEG and EMG electrodes ($n = 6$). Analgesics were administered immediately before surgery (5 mg/kg Metacam and 0.1 mg/kg Vetergesic, subcutaneous) and for at least three days following surgery (0.07 mL Metacam, oral, mixed in 3 mL of jelly). Mice were kept in individual, ventilated cages and monitored at least twice a day until baseline levels of well-being were scored for three consecutive days. They were then moved to their home Plexiglass cages for a week of habituation.

### Pupil diameter monitoring
To monitor levels of arousal, we built a device called an oculometer consisting of an aluminium holder with a miniature camera directed at the eye (NE2D B\&W V90F2.7\_2m, NanEye, Austria) and a far-red LED (Luxeon Rebel LED 720 nm LXML-PF-1, Luxeon Star Leds, Canada) (Supplementary Fig. 7.d), maintained together with Sugru mouldable silicone rubber (Black, Sugru, Germany). The LED was powered through a 700 mA BuckPuck DC driver (3023-D-E-700, Luxeon Star Leds) and connected to the BioAmp Processor (RZ 2, Tucker-Davis Technologies) with an AV 9-way cable (6126423, RS Components) for activation through user input. The camera was connected to a PC via a wired board (985-NanoUSB 2.2, Mouser UK) and the video was recorded with the NanEye Viewer software (NanEye, v6.0.3, default configuration) for NanEye 2D and NanoUSB2 boards at a frame rate between 45 frames per second (fps) and 55 fps. Exposure, gain, and offset were automatically calibrated through the software to minimise loss of data due to poor lighting conditions, which can

sometimes occur during some behaviours such as digging or nest building. The device was secured to the head of the mouse with a slot cheese nylon machine screw (M2 x 6 mm, DIN 84, RS Components).

To test the oculometer, the first recorded images were performed under anaesthesia (n = 3). The videos of freely moving mice (n = 12) were recorded following one 1-hour session of habituation to the device during which all mice were fitted with their respective oculometer, and the arms of the oculometer angled to record the most optimal view of the eye. During recording days, all animals wore the device at least 10 minutes before the start of experiments. The recordings started at light onset and were followed by the injections of either vehicle or 5-MeO-DMT as described below. As the device is quite cumbersome and could impact normal behaviour of the animals, it was decided to only record 60 min of video per subject per session.

## Injections
To investigate the effects of psychedelics not only on vigilance states but also on behaviour and arousal, we used 5-MeO-DMT supplied by Beckley Psytech (BPL-003, Beckley Psytech, UK). 5-MeO-DMT was selected for its strong, fast-acting properties, inducing dream-like experiences in humans and behavioural changes in rodents within 5 min of administration. Following an i.p injection in mice, 5-MeO-DMT has a maximum concentration in plasma after 5 min and a terminal half-life of 12–19 min[55]. On the day of the injection, 1 mg of 5-MeO-DMT was diluted into 1 mL of 0.9% sterile saline, and syringes containing either enough for an injection of 5 mg/kg 5-MeO-DMT or the vehicle (5 mL/kg sterile 0.9% saline) were prepared. The dose of 5-MeO-DMT was selected based on previous work reporting increased head twitch responses at this dose[51,57].

After one day of baseline electrophysiological recording, the mice were subjected to a counterbalanced, crossover design whereby they received a first intraperitoneal injection at light onset (Zeitgeber time (ZT) = 0) with either 5-MeO-DMT or vehicle and were then left undisturbed unless mentioned otherwise (Supplementary Fig. 7.e). Mice were randomly allocated to a specific experimental group. 72 h following the first injections, the mice received an intraperitoneal injection of the substance they did not originally receive (Supplementary Fig. 7.e). The majority of injected animals (n = 26) followed the above injection protocol. A similar version with the injections at a different ZT was performed for another group of animals (n = 16).

To investigate the contribution of 5-HT$_{1A}$ receptors, a group of mice (n = 10) were injected with 1 mg/kg i.p. 5-HT$_{1A}$ receptor antagonist WAY-100635, following previously established protocols[57].

The mice implanted with cannulae (n = 4) received an intracortical injection of 5-MeO-DMT with a total volume of 500nL, at the rate of 50nL per minute. The cannulae were connected to Hamilton syringes (Microliter Syringe 700 series, 22 gauge 80621, Hamilton Company) with a connector cannula (0/SPR 40 cm, Bilaney Consultants Ltd) and thin PE50 clear vinyl tubing (C232CT, Bilaney Consultants Ltd). The infusion rate of the Hamilton syringe was controlled by an automated pump (Harvard Apparatus Pump Controller, Harvard Apparatus), connected to the infuser (Nanomite Injector, Harvard Apparatus). As this method was novel, a concentration of 4.3 mg/mL was used based on previous work in rats using DOI and comparing the ED50 of DOI to the ED50 of 5-MeO-DMT in rats (Willins & Meltzer, 1997).

## Behavioural paradigm
To assess the effects of the compound on exploratory behaviour and feeding, animals (n = 4) received a bowl of 12 sugar pellets (Sucrose Tab/Fruit Punch 14MG, 5TUT - 1811324, TestDiet). On day 0, the bowl was introduced without prior habituation 10 min after dark onset (ZT12). On day 1, the animals received either a vehicle (saline) or a 5 mg/kg 5-MeO-DMT (1 mg/ml saline) injection at dark onset (ZT = 12) at random, followed by two days of recovery, then the other injection on day 4. The bowl of sugar pellets was placed on the cage floor 10 min after the injections. Video recording was acquired for one hour following the injection and analysed offline. All

latencies are expressed from the time the bowl is deposited in the environment and the hand of the experimenter has left the environment.

To investigate the effects of 5-MeO-DMT on behaviour, animals (n = 4) were given access to running wheels as an assay for locomotor activity. Installation of the setup and habituation was performed based on established protocols[108]. In a crossover design, the animals received either vehicle (saline) or a 5 mg/kg 5-MeO-DMT (1 mg/ml saline) injection at dark onset (ZT12) at random on day 1, followed by two days of recovery, then the other injection on day 4. The animals were placed in the running wheels 10 minutes following each injection and were left otherwise undisturbed to assess spontaneous activity. Video recording was acquired for one hour following the injection and analysed offline.

## Sleep deprivation
To investigate the interaction between sleep-wake history and the effects of 5-MeO-DMT, mice (n = 8) were subjected to sleep-deprivation from ZT0 to ZT4 (Supplementary Fig. 7.f). At light onset, the chambers were opened and nesting material removed from the home cages. Removing the nesting material introduces changes to the environment, which kept the animals awake for around 30 min, during which the animals explored their environment or exhibited nest-building behaviours using the bedding of the cage. Repeated nest-building behaviour was immediately followed by a manual introduction of novel objects in the cage. Novel objects included small wooden blocks, folded gloves, and various toys, which were added to the cage or replaced as required. This procedure is known to increase homeostatic sleep pressure and subsequent EEG slow-wave activity during subsequent sleep and relies on the natural exploratory behaviour of mice[103,109,110]. The mice were continuously monitored by the experimenters, and novel objects were introduced into the environment as soon as the animal was immobile or as slow waves became visible on the EEG.

## Data acquisition
Electrophysiological signals were acquired using a multichannel neurophysiology recording system (Tucker-Davis Technologies Inc., Florida, USA). Signals were acquired and processed online using the software package Synapse (Tucker-Davis Technologies Inc., Florida, USA). All signals were amplified (PZ5 NeuroDigitizer preamplifier, Tucker-Davis Technologies Inc., Florida, USA), filtered online (0.1–128 Hz) and stored with a sampling rate of 305 Hz. Signals were read into MATLAB (MathWorks) and filtered with a zero-phase Chebyshev type filter (0.5–120 Hz for EEG, 10–45 Hz for EMG), then resampled at 256 Hz. Exclusion criteria were based on a derivation basis after visual estimation of the signal quality. All analyses were done blind to the treatment.

Vigilance states were manually scored using the signals in European Data Format using SleepSign (Kissei Comtec, Nagano, Japan). The signals were divided into epochs of 4 s. Each epoch was scored based on visual inspections of frontal, occipital, and EMG signals (Supplementary Fig. 7.g). If recorded, LFP signals were also used. Epochs with recording artefacts due to gross movements, chewing, or external electrostatic noise were assigned to the respective vigilance state but not included in the electrophysiological analysis. Wake was scored in epochs containing high EMG activity. NREM sleep was scored in epochs containing low EMG and visual slow activity (0.5 Hz to 4 Hz) of high amplitude in the frontal or in both frontal and occipital derivations. REM sleep was scored when the occipital signal was mostly defined by theta activity (7 Hz to 12.5 Hz) with low EMG. Epochs containing mixed frequencies of lower amplitude and high EMG between two sleep epochs (NREM–NREM or REM–NREM) were scored as micro-arousals, unless 5 or more consecutive similar epochs were scored, in which case it was classified as wake. After sleep scoring, the power spectra of the signal were computed using a fast Fourier transform routine (Hanning window) with a 0.25 Hz resolution and exported in the frequency range between 0 and 120 Hz for spectral analysis.

Fast Fourier transform signals were stored into .mat files using a custom MATLAB script containing the spectrum of each epoch and the associated vigilance state. The spectral extraction script also corrected the

epoch mistakenly scored as micro-arousal when they should have been scored as wake, and vice versa.

For the analysis of pupil diameter video recordings were first converted into AVI format and analysed using DeepLabCut (Mathis et al., 2018; Nath et al., 2019). The model trained for scoring the pupil diameter used the following parameters: resnet = 50, Shuffle 6; 5,000,000 trials. Four points were used to track the pupils: the northernmost, the southernmost, the easternmost and westernmost points (Fig. 3.b). Each point was marked only if it was clearly in view and not hidden by the eyelid or any light artefact. Once scored, the DeepLabCut model was trained on a randomly generated sample of images. The trained model then automatically scored all remaining videos, and only points labelled by the model with a confidence above 99% were kept. Pupil diameter was estimated as the average of the distance between the Northern and Southern point and the distance between the Eastern and Western point.

The time course of vigilance states is always analysed in equivalent time intervals between the drug and vehicle conditions. This enables comparison to previous studies and allows us to take into account the relatively fast kinetics of the substance, which mostly affects vigilance states in the initial intervals, naturally resulting in different amounts of time spent in a specific state early on after drug administration. However, we always ensured that, for spectral analyses, appropriate time intervals were chosen to ensure that all animals contributed to the comparisons with a representative number of artefact-free epochs.

## Data processing

The aperiodic analysis was performed on fast Fourier transform (FFT)–derived EEG power spectra for NREM sleep, baseline wake, and wake following an injection. Spectral slopes were calculated in linear–log space by fitting a straight line to the spectral power between 20 and 120 Hz, in order to minimise the influence of state-specific oscillatory activities prominent during waking and sleep at lower frequencies, such as sigma (10–15 Hz), theta (6–9 Hz), and delta (0.5–4 Hz). To assess changes in cross-frequency coupling, we calculated the modulation index for phase frequencies between 1.5 Hz and 10.5 Hz and amplitude frequencies between 0.25 Hz and 100.5 Hz using previously established MATLAB toolboxes[111].To investigate the incidence of local field potential (LFP) or EEG slow waves and corresponding neuronal activity, LFP signals in $n = 4$ mice were analysed as previously[38]. Briefly, the LFP signal was first bandpass filtered between 0.5–4 Hz (stopband edge frequencies 0.2–8 Hz) with MATLAB filtfilt function exploiting a Chebyshev Type II filter design, and waves were detected as positive deflections of the filtered LFP signal between two consecutive negative deflections below the zero-crossing. Only LFP waves with a peak amplitude larger than the mean plus two standards deviations of the amplitude across all detected waves were included in MUA analyses, while all waves were utilised for plotting the distribution of amplitudes and durations. For MUA analysis, slow waves were aligned to their positive peak, and the corresponding average profile of neuronal spiking was computed (Supplementary Fig. 7.h-i).

## Statistics and reproducibility

Statistical analyses were computed using Prism (GraphPad Software, Boston, Massachusetts USA, version 10.2.0 (392)). All data sets were tested for normality with the Shapiro-Wilk test, and sphericity tested with Spearman's rank correlation test. If all assumptions were met, a parametric test was conducted; otherwise, its non-parametric equivalent was selected for $\alpha = 0.05$. Each figure will detail which tests were used. On figures, a significant difference is plotted as a straight black or coloured line in spectrogram analyses, as plotting the difference for each data bin would make the figure less readable. For all other figures, significance levels are indicated with black asterisks: * for $p < [0.05; 0.01]$, ** for $p < [0.01; 0.001]$, *** for $p < [0.001; 0.0001]$, **** for $p < 0.0001$. Figures were generated using Prism and show group means with SEM for parametric data, unless mentioned otherwise in the figure legend, such as for plotting representative raw signals of one animal. All boxplots show the median, interquartile ranges, and the minimum and maximum values.

## Reporting summary

Further information on research design is available in the Nature Portfolio Reporting Summary linked to this article.

## Data availability

All data presented in this manuscript are available on figshare (10.6084/m9.figshare.30058069). All raw data are available upon reasonable request to the corresponding authors.

## Code availability

All codes are available upon reasonable request to the corresponding authors.

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

## Acknowledgements

The authors would like to thank Laura McKillop, Christian Harding, Elise Meijer and Sian Wilcox for their assistance with surgery, drug preparation, equipment setup, animal husbandry, or data analysis; Gianina Ungurean and Niels Rattenborg for advice on ordering and installation of miniature cameras for monitoring pupil diameter; Stuart Peirson and Carina Pothecary on the materials and review of the oculometer; Vanda Reiss for her drawing of a mouse wearing the occulometer (Fig. 4.a); Matt Jones for his valuable comments; and Beckley Psytech for providing the compound for this study. The study was funded by UK Biotechnology and Biological Sciences Research Council grant (BB/M011224/1); Wellcome Trust Senior Investigator Award (106174/Z/14/Z); Wellcome Trust Strategic Award (098461/Z/12/Z); John Fell OUP Research Fund Grant (131/032); and the Medical Research Council (MR/S01134X/1).

## Author contributions

B.J.B.B.: Conceptualisation; Methodology, Investigation, Visualisation, Funding acquisition, Writing (original draft, review and editing). J.P.M.: Methodology, Visualisation, Writing (review and editing). A.A.: Methodology, Visualisation, Writing (review and editing). A.H.S.: Methodology, Visualisation, Writing (review and editing). J.P.: Methodology, Visualisation, Writing (review and editing). D.M.B.: Conceptualisation, Supervision, Writing (review and editing). T.S.: Conceptualisation, Supervision, Writing (review and editing). V.V.V.: Conceptualisation, Investigation, Funding acquisition, Supervision, Writing (original draft, review and editing)

## Competing interests

The authors declare no competing interests.
