## [Transparent Peer Review file · Communications Biology]

Vigilance state dissociation induced by 5-MeO-DMT in mice

Corresponding Author: Professor Vladyslav Vyazovskiy

Version 0:

Reviewer comments:

Reviewer #1

(Remarks to the Author)

The manuscript "Vigilance state dissociation induced by 5-MeO DMT in mice" by Benjamin J B Bréant et al. explores the effects of an injection of a short-acting psychedelic 5-methoxy-N,N-dimethyltryptamine (5-MeO-DMT) on freely moving adult male mice by performing chronic electrophysiological recordings from the neocortex concomitant with pupillometry. Results show that 5-MeO-DMT induced an acute dissociated state of arousal with prominent slow oscillations in the cortex accompanied with theta frequencies suppression and wake-like levels of fast frequencies. They also found a marked pupil dilation in behaviorally awake, moving animals. They found a reduction of REM sleep as well that was overcompensated in the subsequent 48 hours. They further studied the potential mechanisms that induced those slow waves. They present novel results for an interesting issue. The methodology and analysis are well suited for this study. The conclusions are correct but the discussion is somewhat brief.

This reviewer recommends enriching the discussion by comparing their results with 5-MeO DMT against other drugs that also induce a dissociated state of arousal with NREM sleep like slow oscillations and wake-like levels of fast frequencies. For example, acetylcholine muscarinic antagonists like atropine and scopolamine, which are not mentioned by the authors, produce an arousal-dissociated state that has some similitudes with 5-MeO-DMT.

- It produce a dissociated state with wakefulness behavior and slow waves (Wikler, 1952) PMID: 14912089 DOI: 10.3181/00379727-79-19345 (Frohlich et al. 2023) <https://doi.org/10.1038/s42003-023-04988-8>
- The neocortical recordings are characterized with NREM like slow waves accompanied with wake like gamma oscillations (Castro-Zaballa et al.2019) <https://doi.org/10.1016/j.bbr.2018.10.016>, (Frohlich et al. 2023) <https://doi.org/10.1038/s42003-023-04988-8>
- This slow waves also has negative "up states" with gamma activity and positive "off periods" without it. (Castro-Zaballa et al.2019) <https://doi.org/10.1016/j.bbr.2018.10.016>, (Frohlich et al. 2023) <https://doi.org/10.1038/s42003-023-04988-8>
- It also produces a marked pupil dilation in behaviourally awake, moving animals. (Wikler, 1952) PMID: 14912089 DOI: 10.3181/00379727-79-19345 (Frohlich et al. 2023) <https://doi.org/10.1038/s42003-023-04988-8>

Scopolamine and atropine have some differences like the presence of NREM sleep like "sleep spindles" during the wake like state with slow waves and fast activity. The authors do not describe the presence of spindles during 5-MeO DMT effect. Do you find spindles like activity? It would be interesting that the authors describe the absence or presence of sleep spindles under 5-MeO DMT effect. Ketamine in subanesthetic doses also produce a kind of slow waves that are different than the NREM sleep ones accompanied with fast activity that is different than the one of wakefulness (and sleep).

Reviewer #2

(Remarks to the Author)

In this manuscript, Breant et al. present a thorough electrophysiological and behavioral characterization of a drug-induced state in mice following administration of 5-MeO-DMT. In the process, they designed a custom-made, low-cost system, the "oculometer," to reliably record the animal's pupil in freely moving mice without changing their wake EEG patterns. They find that brain activity during this state is marked by slow-wave EEG activity associated with neuronal off-states, similar to those seen in NREM sleep, superimposed on high-frequency activity typically associated with wakefulness. They also find evidence of behavioral and autonomic arousal in this state, including typical waking movements and pupil dilation, and a short-term suppression of REM sleep with a subsequent rebound. Based on these observations, they argue that 5-MeO-DMT induces a dissociated state, combining features of both sleep and wakefulness. They further hypothesize that the presence of sleep-like processes may help explain certain psychedelic effects, such as dream-like hallucinations and

enhanced plasticity.

Overall, this is a well-conducted study on a novel, valuable, and rich dataset that represents a significant step forward in our understanding of the neuronal mechanisms underlying altered states of consciousness induced by psychedelics, and their relation to sleep physiology. The central claim that 5-MeO-DMT induces a dissociated state resembling both sleep and wakefulness is very interesting, but appears only partially supported by the results. Several methodological aspects would need clarification to strengthen the overall conclusions.

Major comments:

1) The central claim—that 5-MeO-DMT induces a dissociated state combining elements of both sleep and wakefulness—largely hinges on the observation of EEG slow waves and their association with neuronal off-states. A homeostatic regulation of the EEG slow waves could not clearly be shown. However, other key features that would bolster a sleep-like interpretation are either not reported or insufficiently explored. For instance:

- a. Did the authors detect other typical NREM sleep features such as sleep spindles, hippocampal sharp-wave ripples, or infraslow fluctuations in sigma power?
- b. Was there any evidence of sensory disconnection? The observation that animals are slower to interact with objects (e.g., a cup or pellets) is not convincing as an argument, as it may reflect other factors like reduced motivation. The fact that the animals seemingly interacted with their environment argues against a total sensory disconnection in the sense of a functional blindness, as is sometimes observed in sleep-related dissociated states.
- c. Concerning the homeostatic regulation of sleep. Was the typical decay in delta power during subsequent sleep episodes altered following 5-MeO-DMT administration compared to vehicle controls? A reduced delta power decay would suggest that the drug-induced state contributes to sleep pressure dissipation, strengthening the case for its sleep-like nature.

If such supporting evidence is unavailable, it may be advisable to temper the strength of the main claim. A more descriptive framing—highlighting similarities to sleep without asserting a true dissociative state—would preserve the value of the findings while aligning more closely with the data. Importantly, such a revision would not detract from the significance of the study, which remains commendable.

2) The discussion makes several claims about neural plasticity, particularly in connection to sleep-like mechanisms in association with 5-MeO-DMT. However, plasticity was not directly investigated in this study, and the purported link to sleep features remains speculative. These interpretations would benefit from a more cautious tone or explicit framing as hypotheses for future investigation. Similarly, the statement “we surmise that the 5-MeO-DMT induced slow waves have a distinct mechanism from bona fide sleep slow waves” seems not entirely justified by the findings, because 5-MeO-DMT-associated slow waves did show the off-periods as natural slow waves during NREM sleep.

3) Were other important features of sleep architecture and microarchitecture, such as microarousal density and dynamics, and NREM and REM sleep bout lengths assessed following the drug administration?

4) The experiment using the 5-HT_{1A} antagonist is highly relevant to understanding serotonergic mechanisms of sleep. Did the authors explore the sleep features after these manipulations? Additionally, did the authors analyze the behavioral changes of the animals in this condition?

5) All power dynamics (e.g., Fig. 3c and similar) should be calculated relative to the vigilance state from which it is derived (Franken et al., 1991; Fernandez et al., 2018; Hubbard et al., 2020, Figure 1). For instance, to calculate homeostatic Delta decay during sleep, it should be done by dividing the recording into bins of equal amount of NREM; Then, the average delta power for each bin is computed, including only the NREM epochs. A second normalization to the values between ZT8-12, to the first hour, or similar is also commonly used (Fernandez et al., 2018 Sup Figure 4b). Otherwise, the dynamics might reflect or be contaminated by differences in sleep architecture. In fact, this might be the reason why the authors do not observe the well described “SWA” decay after sleep deprivation or from the beginning of sleep.

6) As theta activity during wake has been related to exploration, and because the animals explored less after DMT, it raises the question of how much the reduction in theta is related to changes in the animal's behavior. One alternative to check this is to correlate the changes in the theta to exploration or behavioral metrics during the post-injection period.

7) One particularly intriguing observation is the presence of fast-frequency activity superimposed on slow waves, as seen, for example, in Figure 1b. This pattern is reminiscent of EEG features observed in human sleep under the influence of certain drugs, and of some slow waves contained in micro-arousals in sleep. The authors are uniquely well-positioned to investigate and interpret these dynamics. Do these patterns reflect asynchronous activity across distinct neuronal populations—some exhibiting wake-like activity while others display sleep-like slow waves? Or might they originate from the same neuronal populations, perhaps reflecting increased firing during the upstate of the slow oscillation? What are the features of the fast activity? Is there any oscillatory (periodic) component associated with it? Which range of frequencies does it comprise? Is it only a reflection of the aperiodic components? The quantification and comparison of the MUA of NREM sleep slow waves and those related to the drug might also shed light on these matters in any case, an in-depth discussion and analysis of such ‘ultralocal’ aspects of sleep underlying the findings would be valuable here.

8) What was the rationale for studying the overall SWA instead of the two components of this slow activity? Cortico-cortical Slow Oscillations (SO) and thalamo-cortical delta activity (Steriade 1994) have different dynamics across sleep, respond differently to sleep pressure (Hubbard et al., 2020), and thalamic manipulations (Fernandez et al., 2018; Osorio-Forero, et

al., 2021). Furthermore, the local aspects of different slow waves might also differ as in humans (Siclari et al., 2014; Navarrete et al., 2023). Using at least two distinct bands would help strengthen the interpretation and claims of the study. Addressing the differences between SO and Delta is also essential for the interpretation of the fast-frequency activity superimposed on the slow waves and the experiments involving local 5-MeO-DMT infusion in the cortex.

9) When interpreting the regional findings (anterior vs. posterior findings), the authors may consider the regional distribution of serotonin. In the human cortex, serotonin metabolites follow a clear posterior-anterior gradient, highest in posterior cortical regions (Javoy Agid et al. 1989).

Minor comments

- 10) "State of arousal" in the introduction is an unusual term in this context. Consider using "behavioral state" or "vigilance state" for clarity.
- 11) The term "intoxication" in the second paragraph of the introduction may imply overdose or pathology. Consider replacing it with "influence" or "administration" to avoid misleading connotations.
- 12) Some information in the main text or the figures is missing, such as the sleep stage analyzed, for instance, in the slow waves analysis.
- 13) Concerning the non-oscillatory analysis, what is the rationale behind using this band? This analysis is not described in the methods, and differences in sleep stages are also not addressed across the groups.
- 14) For the cumulative time spent in REM, it is recommended to show the relative changes instead of the absolute cumulative time (McCarthy et al., 2016), as they might help to differentiate whether REM sleep homeostasis was altered or just shifted. This might be useful to address topics such as the one raised in: "One possibility is that the dissociated state induced by 5-MeO-DMT could replace some aspect(s) or function(s) of REM sleep and thus reduce the homeostatic pressure to express REM sleep."
- 15) For the MUA analysis, it is recommended to use a raster plot or a moving average, such as in Aston-Jones and Bloom, 2981 or Mann and Paulsen (2010).
- 16) In (Fig. 1.f-h-l, Movie S2), there is no l. (Fig. 1.j; Fig. S4.a, Movie S2), There is no "j".
- 17) The movies are a highly valuable information. Merging Movies S1 and S2, showing the animal's behavior and concomitant signals, could be more informative. Otherwise, Movie S2 does not add further information to Figure 1a-b, and it would be more convincing to show that the slow waves appear simultaneously as the animal is moving. Also, the file of the last movie is broken or empty.
- 18) The panels 1a-b and 1f-i might be redundant (with further examples in the supplementary data S7). For space purposes, including a complete description of the sleep stages with the LFP, EEG, EMG, and MUA activity might be sufficient.
- 19) Concerning the methods
 - a. To support the observations, it is recommended that example images of the location of the cannulas and LFP electrodes be used, particularly for the experiment on local changes. Ideally for all animals, but at least a couple of examples.
 - b. What was the dosage of the oral Metacam?
 - c. The "Epochs containing mixed frequencies of lower amplitude and high EMG between two sleep epochs (NREM–NREM or REM–NREM) were scored as movement" of < 12 - 16 seconds are usually studied as Microarousals (except for the REM–NREM transition), is there any reason why this nomenclature was not used, also, what was the rationale not to study changes in MAs in the study, particularly given that some discussion and conclusions were related to arousal.
 - d. Concerning the slow-wave detection, it would be crucial to show examples of the detected waves. Similarly, did using a standard-deviation-based threshold change the type of waves that could have been detected? Which kind of slow waves were detected during wake without DMT? Why were no additional time features, such as time between 0 crosses, used for the detected waves?
 - e. No methodological description of the aperiodic analysis was provided.
 - f. The authors did not show differences in behavior, sleep, etc., in animals with the oculometer compared to animals without the system.
- 20) In regard to the figures, the idea of providing a result description for every panel might be confusing unless defined by the journal; it is not encouraged. Concerning specific figures,
 - a. Figure 1b: Please clarify whether the NREM and REM traces reflect natural sleep or sleep occurring under the influence of the drug. Also, specify the polarity convention (i.e., is negative up or down?). The last sentence, "The mean of all non-representative data is plotted with SEM," only refers to d and e?
 - b. Figure 2: Is the vehicle in b and c with or without SD?
 - c. Figure 4: The title "The effects of 5-MeO-DMT are mediated by 5-HT1A receptors." is misleading. The results show that some effects are mediated by these receptors, but others are not. In this figure, it is also recommended to homogenize the panel dimensions. As mentioned before for Figure 3, it is essential to control for sleep stages when describing dynamic power changes. The result "The suppression of the occipital theta activity occurring after the injection of 5-MeO-DMT was largely suppressed by an injection of WAY resulting in vehicle-like levels of theta activity throughout interest (ME analysis, effect of time point*condition F87,143 = 2.92, $p < 0.0001$; Fisher's LSD $p > 0.05$)." is not commented in the manuscript.
 - d. Figure S1: The NREM trace appears unusually jagged—could this be an artifact? If not, some explanation would be helpful.
 - e. Figure S3: "Latency is expressed from the time the bowl is deposited in the home cage and the hand of the experimenter is removed from the environment" might better be described in the methods. What is the n for each experiment? (it is recommended to add it for each panel, or group of experiments) And why is there no difference in f if the values seem quite different? It is recommended to show each data point in the plots.
 - f. Figure S4: It is recommended to show the grand average for all animals in addition to the representative animal, a longer window around the peak of the waves, and representative examples of single waves in the three conditions (BL NREM, vehicle wake, and 5-MeO-DMT). What is the count, and not only the percentage of waves? Did they differ in number? They

seem very different, in regarding the wake condition in the vehicle.

g. Figure S5: It is recommended to show the wake and the sleep-related FFT. Also for Figure 1.

h. Figure S7: There is no 2.2 in the figure. As mentioned, "e" might be redundant; you already have examples in Fig. 1. Additionally, add the FFT of all sleep stages for all manipulations instead.

References

- Steriade, M. (1994). Sleep oscillations and their blockage by activating systems. *Journal of Psychiatry and Neuroscience*, 19(5), 354.
- Hubbard, J., Gent, T. C., Hoekstra, M. M., Emmenegger, Y., Mongrain, V., Landolt, H. P., ... & Franken, P. (2020). Rapid fast-delta decay following prolonged wakefulness marks a phase of wake-inertia in NREM sleep. *Nature communications*, 11(1), 3130.
- Fernandez, L. M., Vantomme, G., Osorio-Forero, A., Cardis, R., Béard, E., & Lüthi, A. (2018). Thalamic reticular control of local sleep in mouse sensory cortex. *Elife*, 7, e39111.
- Navarrete, M., Osorio-Forero, A., Gómez, A., Henao, D., Segura-Quijano, F. E., Le Van Quyen, M., & Valderrama, M. (2023). Response of sleep slow oscillations to acoustic stimulation is evidenced by distinctive synchronization processes. *Sleep*, 46(6), zsad110.
- Siclari, F., Bernardi, G., Riedner, B. A., LaRocque, J. J., Benca, R. M., & Tononi, G. (2014). Two distinct synchronization processes in the transition to sleep: a high-density electroencephalographic study. *Sleep*, 37(10), 1621-1637.
- Franken, P., Dijk, D. J., Tobler, I., & Borbély, A. A. (1991). Sleep deprivation in rats: effects on EEG power spectra, vigilance states, and cortical temperature. *American Journal of Physiology-Regulatory, Integrative and Comparative Physiology*, 261(1), R198-R208.
- Javoy-Agid, F., Scatton, B., Ruberg, M., L'heureux, R., Cervera, P., Raisman, R., ... & Agid, Y. (1989). Distribution of monoaminergic, cholinergic, and GABAergic markers in the human cerebral cortex. *Neuroscience*, 29(2), 251-259.
- Osorio-Forero, A., Cardis, R., Vantomme, G., Guillaume-Gentil, A., Katsioudi, G., Devenoges, C., ... & Lüthi, A. (2021). Noradrenergic circuit control of non-REM sleep substates. *Current Biology*, 31(22), 5009-5023.
- McCarthy, A., Wafford, K., Shanks, E., Ligocki, M., Edgar, D. M., & Dijk, D. J. (2016). REM sleep homeostasis in the absence of REM sleep: effects of antidepressants. *Neuropharmacology*, 108, 415-425.
- Aston-Jones, G., & Bloom, F. (1981). Activity of norepinephrine-containing locus coeruleus neurons in behaving rats anticipates fluctuations in the sleep-waking cycle. *Journal of Neuroscience*, 1(8), 876-886.
- Mann, E. O., & Paulsen, O. (2010). Local field potential oscillations as a cortical soliloquy. *Neuron*, 67(1), 3-5.

Reviewer #3

(Remarks to the Author)

The manuscript reported that the injection of serotonin psychedelic, non-specific 5-HT receptor agonist 5-MeO-DMT induced a transient dissociated state of vigilance, characterized by coexistence of sleep-like slow oscillations in the brain signal and marked pupil dilation in freely behaving awake mice. The authors employed multiple approaches, including electrophysiological vigilance state recordings, LFP, pupil diameter monitoring using custom-built oculometer, power spectral slope analysis, behavioral video recording, intracortical injection. The study design is logical and well-structured and supported the previous reports using 5-MeO-DMT (Bréant, B. J. B. et al. *Sleep Med.* 100, S22 (2022).; Souza, A. C. et al. *Sci. Rep.* 14, 11281 (2024)).

1- While slow waves appear on the EEG following 5-MeO-DMT administration, they are clearly different than one during slow wave sleep. Fig. 1i, the slow waves seem to have a higher frequency, and a lower amplitude as compared to SWS (Fig. 1g). It would be useful to have a direct comparison between SWS and post 5-MeO-DMT FFT analysis.

2- The use of a single 5 mg/kg dose of 5-MeO-DMT is justified based on prior studies reporting increased head twitch response at this concentration. However, incorporating multiple doses to assess dose-dependency would significantly strengthen the understanding of 5-MeO-DMT-generated slow waves.

3- "Sleep" should always follow REM and NREM in the text, figures and figure captions. If an abbreviation is needed, the abbreviation for REM sleep should be REMS and the abbreviation for NREM sleep should be NREMS. REM stands for rapid eye movement which is not a sleep stage.

4- Video 2: The EEG seems to not be detected during NREM sleep and REM sleep, only a line is seen. On the opposite, during wakefulness and post 5-MeO-DMT injection, the EEG fluctuations are clearly seen. REM sleep is labeled as PS and NREM sleep is labeled as SWS, which can be confusing for non-sleep experts. The manuscript is using REM sleep and NREM sleep, the video should keep these denominations.

5- Regarding the Supplementary Videos demonstrating the coexistence of active behaviors and slow oscillations in brain signals, would it be possible to integrate the waking behavior along with the corresponding EEG and LFP signals from a single mouse into one synchronized video? This could provide a more intuitive understanding of the temporal relationship between behavior and neural activity

6- Fig. S3f-g: Statistical analysis did not show a significant difference in average speed and maximum speed. However, the charts show a trend to decrease. The video S4 also indicates that the mice are spending less time running the wheel. According to the method section, only 4 mice were included in the study. This number is probably too low. Interestingly, these parameters are also affected by the control injection, indicating that animal handling and the pain from the injection affect these behaviors. We can wonder if the protocol of the motor test is appropriate. In fact, the sample size (n=4 per group)

is small for all behavioral paradigms assessing exploratory behavior, feeding, and locomotor activity. Increasing the number of animals would enhance the statistical power of the findings.

7- To address the origin of the induction of SWA, the authors did not observe any marked behavioral or EEG alterations by intracortical infusion of 5-MeO-DMT in the primary somatosensory cortex and argue that the effects on brain state and arousal are more likely to be generated at the global level. Please clarify why the prefrontal cortex was not the target of choice given that slow waves are originating from the prefrontal cortex.

8- Co-treatment with 1 mg/kg of the 5-HT_{1A} receptor antagonist WAY did not counteract some of the effects of 5-MeO-DMT and instead further increased SWA. Please clarify the rationale for selecting the 1 mg/kg dosage of WAY. As WAY is not a selective 5-HT_{1A} receptor antagonist, the authors should discuss the possible use of 5-HT_{2A-R} and 5-HT_{1A-R} knockout mice.

9- Some descriptions in the Results section appear to rely on qualitative observations rather than quantitative statistical analysis. Incorporating appropriate quantitative analyses would improve the scientific rigor of the findings.

For example: Page 3

Plotting the EEG spectrograms following vehicle and 5-MeO-DMT revealed that marked effects of the compound on cortical activity lasted less than an hour in all cases (representative example: Fig. 1.c). Specifically, we observed that while the animals were awake and moving after the injection of 5-MeO-DMT, theta-frequency activity was replaced by slow frequencies for approximately 45 min before returning to levels comparable to vehicle (Fig. 1.c; Movie S2)

10- Fig. S4b-c: The figure caption indicates that NREM sleep and wakefulness after the injection of vehicle or 5-MeO-DMT are presented but the chart legend indicates "Baseline NREM, Vehicle and 5-MeO-DMT".

11- The analysis of power spectral slopes in the 20–30 Hz gamma band across conditions is intriguing. However, it would be helpful to clarify why only studies this specific frequency range.

12- It will be useful to have the number of animals included in each experiment in the figure captions.

13- The figure order can be optimized so they are cited in order in the text.

14- Please include the full term for this abbreviation at first use:

5-HT_{2A/2c} agonist 1-(2,5-dimethoxy-4-iodophenyl)-2-aminopropane (DOI)

Reviewer #4

(Remarks to the Author)

Version 1:

Reviewer comments:

Reviewer #1

(Remarks to the Author)

The authors have adequately answered to my comments and addressed my concerns. I have no more suggestions to make.

Reviewer #2

(Remarks to the Author)

In this updated version of the manuscript, Breant et al. provided thoughtful responses to the reviewers' comments. Particularly, the additional analyses of sleep homeostasis (including the NREM SWA rebound following SD+5-MeO-DMT) and expanded discussion have strengthened the manuscript and its conclusions. Once again, it is worthwhile to highlight the value of the dataset derived from this experimental work.

Overall, the study now presents a more balanced and compelling account of the 5-MeO-DMT-induced state, appropriately framed as a different rather than sleep-like state. Only a few remarks are worth mentioning at this point, including:

Why did you choose 20-120 Hz for the aperiodic activity? Additionally, do you fit in the linear-linear space, the Linear-Log Space, or the Log-Log space? The first one is uncommon and may be the reason why the values differ significantly from those in the literature.

Related to the experiment of local injection of DMT, in contrast to the author's conclusion, "These results suggest that the prominent effects on the brain state and arousal are more likely to be generated at the global level". Another possibility is that it is local but in a subcortical (or another cortical) area. The results only suggest that it's not in the local S1 vicinity. A good example of this is the effects of Noradrenaline on spindles (an effect mediated by the thalamic terminals of the LC, but

not the cortical ones).

Reviewer #3

(Remarks to the Author)

The authors have addressed the main concerns from the first submission. The revised manuscript is improved with clearer explanations and stronger evidence supporting the main conclusions.

There are only minor typos:

Supplementary Video 4: SWS and PS are used instead of NREM sleep and REM sleep respectively. That could be confusing to some readers.

Page 5, the sentence "Firstly, we replicated the effects of the injection of 5-MeO-DMT in animals wearing the oculometer (Fig. S6)," should reference Fig S5.

Fig. S7 is a method figure and is not showing "a minor increase in EEG SWA was observed after the infusion (Fig. S7)."
Page 6

Fig. S8 is not cited in the manuscript

Reviewer #4

(Remarks to the Author)

I co-reviewed this manuscript with one of the reviewers who provided the listed reports. This is part of the Communications Biology initiative to facilitate training in peer review and to provide appropriate recognition for Early Career Researchers who co-review manuscripts.

Reviewer #5

(Remarks to the Author)

I co-reviewed this manuscript with one of the reviewers who provided the listed reports. This is part of the Communications Biology initiative to facilitate training in peer review and to provide appropriate recognition for Early Career Researchers who co-review manuscripts.

We thank the reviewers for reading our manuscript carefully, and for providing excellent constructive comments and recommendations that helped us to significantly improve our manuscript.

Please find below our responses to the comments.

Reviewer #1 (Remarks to the Author):

1) *The manuscript "Vigilance state dissociation induced by 5-MeO DMT in mice" by Benjamin J B Bréant et al. explores the effects of an injection of a short-acting psychedelic 5-methoxy-N,N-dimethyltryptamine (5-MeO-DMT) on freely moving adult male mice by performing chronic electrophysiological recordings from the neocortex concomitant with pupillometry. Results show that 5-MeO-DMT induced an acute dissociated state of arousal with prominent slow oscillations in the cortex accompanied with theta frequencies suppression and wake-like levels of fast frequencies. They also found a marked pupil dilation in behaviorally awake, moving animals. They found a reduction of REM sleep as well that was overcompensated in the subsequent 48 hours. They further studied the potential mechanisms that induced those slow waves. They present novel results for an interesting issue. The methodology and analysis are well suited for this study. The conclusions are correct but the discussion is somewhat brief.*

We thank the reviewer for their positive evaluation of our manuscript and for offering advice on how to improve the discussion, which we now addressed as detailed below.

2) *This reviewer recommends enriching the discussion by comparing their results with 5-MeO DMT against other drugs that also induce a dissociated state of arousal with NREM sleep like slow oscillations and wake-like levels of fast frequencies. For example, acetylcholine muscarinic antagonists like atropine and scopolamine, which are not mentioned by the authors, produce an arousal-dissociated state that has some similitudes with 5-MeO-DMT.*

- *It produce a dissociated state with wakefulness behavior and slow waves (Wikler, 1952) PMID: 14912089 DOI: 10.3181/00379727-79-19345 (Frohlich et al.*

- 2023) <https://doi.org/10.1038/s42003-023-04988-8>*

- *The neocortical recordings are characterized with NREM like slow waves accompanied with wake like gamma oscillations (Castro-Zaballa et al.2019) <https://doi.org/10.1016/j.bbr.2018.10.016>, (Frohlich et al.*

- 2023) <https://doi.org/10.1038/s42003-023-04988-8>*

- *This slow waves also has negative "up states" with gamma activity and positive "off periods" without it. (Castro-Zaballa et al.2019) <https://doi.org/10.1016/j.bbr.2018.10.016>, (Frohlich et al. 2023) <https://doi.org/10.1038/s42003-023-04988-8>*

- *It also produces a marked pupil dilation in behaviourally awake, moving animals. (Wikler, 1952) PMID: 14912089 DOI: 10.3181/00379727-79-19345 (Frohlich et al. 2023) <https://doi.org/10.1038/s42003-023-04988-8>*

We apologise for this omission. Indeed, while we outlined a number of conditions where sleep-like slow waves are observed outside of the sleep context, our coverage of drug-induced slow waves was somewhat limited, which we have now addressed. We now included a new paragraph in the discussion to address these important studies (page 8).

3) *Scopolamine and atropine have some differences like the presence of NREM sleep like "sleep spindles" during the wake like state with slow waves and fast activity. The authors do not describe the presence of spindles during 5-MeO DMT effect. Do you find spindles like activity? It would be interesting that the authors describe the absence or presence of sleep spindles under 5-MeO DMT effect. Ketamine in subanesthetic doses also produce a kind of slow waves that are different than the NREM sleep ones accompanied with fast activity that is different than the one of wakefulness (and sleep).*

We thank the reviewer for bringing up the possibility of the occurrence of spindle-activity and faster frequencies during waking after 5-MeO DMT, similar to what has been found after ketamine. Indeed, we were interested in exploring this aspect, and had carefully examined the raw signals in all animals. However, we were never able to visually detect an occurrence of spindle events, which are quite prominent in the signals we typically record, and we have a great deal of experience of detecting and analysing those during sleep (Blanco-Duque et al., 2024 Science Advances). Since no visually detectable spindle activity was present under 5-MeO DMT, and no spectral peak in the spindle-frequency range was observed on the spectra, we do not believe we have strong basis for analysing this further. We now comment on this in the Results section (page 3) and in the Discussion (page 8).

Considering the 2nd point, we have now undertaken more detailed analysis of fast frequencies which revealed that they were overall similar to normal wakefulness (see our response to Reviewer 2 below).

Reviewer #2 (Remarks to the Author):

In this manuscript, Breant et al. present a thorough electrophysiological and behavioral characterization of a drug-induced state in mice following administration of 5-MeO-DMT. In the process, they designed a custom-made, low-cost system, the “oculometer,” to reliably record the animal’s pupil in freely moving mice without changing their wake EEG patterns. They find that brain activity during this state is marked by slow-wave EEG activity associated with neuronal off-states, similar to those seen in NREM sleep, superimposed on high-frequency activity typically associated with wakefulness. They also find evidence of behavioral and autonomic arousal in this state, including typical waking movements and pupil dilation, and a short-term suppression of REM sleep with a subsequent rebound. Based on these observations, they argue that 5-MeO-DMT induces a dissociated state, combining features of both sleep and wakefulness. They further hypothesize that the presence of sleep-like processes may help explain certain psychedelic effects, such as dream-like hallucinations and enhanced plasticity.

Overall, this is a well-conducted study on a novel, valuable, and rich dataset that represents a significant step forward in our understanding of the neuronal mechanisms underlying altered states of consciousness induced by psychedelics, and their relation to sleep physiology. The central claim that 5-MeO-DMT induces a dissociated state resembling both sleep and wakefulness is very interesting, but appears only partially supported by the results. Several methodological aspects would need clarification to strengthen the overall conclusions.

We thank the reviewer for their highly positive evaluation of our manuscript, and for extensive recommendations which helped us to revise the manuscript.

Major comments:

1) The central claim—that 5-MeO-DMT induces a dissociated state combining elements of both sleep and wakefulness—largely hinges on the observation of EEG slow waves and their association with neuronal off-states. A homeostatic regulation of the EEG slow waves could not clearly be shown. However, other key features that would bolster a sleep-like interpretation are either not reported or insufficiently explored. For instance:

a. Did the authors detect other typical NREM sleep features such as sleep spindles, hippocampal sharp-wave ripples, or infraslow fluctuations in sigma power?

b. Was there any evidence of sensory disconnection? The observation that animals are slower to interact with objects (e.g., a cup or pellets) is not convincing as an argument, as it may reflect other factors like reduced motivation. The fact that the animals seemingly

interacted with their environment argues against a total sensory disconnection in the sense of a functional blindness, as is sometimes observed in sleep-related dissociated states.

c. Concerning the homeostatic regulation of sleep. Was the typical decay in delta power during subsequent sleep episodes altered following 5-MeO-DMT administration compared to vehicle controls? A reduced delta power decay would suggest that the drug-induced state contributes to sleep pressure dissipation, strengthening the case for its sleep-like nature.

If such supporting evidence is unavailable, it may be advisable to temper the strength of the main claim. A more descriptive framing—highlighting similarities to sleep without asserting a true dissociative state—would preserve the value of the findings while aligning more closely with the data. Importantly, such a revision would not detract from the significance of the study, which remains commendable.

We thank the reviewer for raising this important point, and we agree that caution is warranted in equating the state induced by 5-MeO DMT or accompanying brain oscillations with sleep. It is precisely for this reason we are largely referring to the state as a dissociated state rather “sleep-like” as it was in the earlier version of our manuscript deposited on Biorxiv last year. We next address the specific points in turn:

a) As explained above in our response to Reviewer 1, who is raising a similar point, we were not able to visually detect any events resembling spindles, nor we have access to recordings from the hippocampus, which would have allowed us to reliably record SWRs. We now briefly mention this in the revised manuscript (page 3, 8), but we also would like to highlight that further experiments may be necessary to conclusively rule out the possibility that these oscillations are present under 5-MeO DMT. We recorded only from a few brain regions, and have no data from the primary somatosensory cortex, where spindle activity, for example, is quite prominent (Blanco-Duque et al., 2024 Science Advance and work from Anita Lüthi lab). Considering fast-frequency activity, we would like to remind that we found no difference between 5-MeO DMT waking and spontaneous wakefulness. This means that fast frequencies were quite prominent under the drug, concomitant with slow waves, more typical for sleep – hence our characterisation of the state as dissociated, having features of both waking and sleep.

b) We completely agree that the behavioural evidence we provide is a rather weak argument for sensory disconnection, and this is why our conclusions are very careful in this regard. We would like to remind that the primary purpose of the behavioural experiments was to rule out that 5-MeO DMT induces an abnormal state, especially related to serotonin syndrome. We did not aim to provide a thorough quantitative analysis of behaviour, but rather aimed to demonstrate the lack of obvious abnormalities in awake state, and this is what we believe the study provides unequivocally. We also agree with the reviewer that the signs of disengagement we observed may indicate anhedonia, but still it was clear that the animals showed awareness of the surrounding and therefore we have no strong evidence for disconnection. We must add that the usual assumption of disconnection during sleep is likewise questionable. There is abundant evidence that the brain is responding to the sensory stimuli from outside during all sleep stages. In the revised manuscript we now made it clear that we do not imply disconnection induced by psychedelics, neither compare 5-MeO induced state with sleep using this criterion.

c) We thank the reviewer for this comment, which prompted us to perform additional analysis. Firstly, we found that SWA during NREM sleep was initially similar between 5-MeO DMT and vehicle when injection was done close to the light onset without sleep deprivation (see Fig. S1.a). However, we now report that SWA during NREM without sleep deprivation following an injection at ZT = 4 was significantly lower than vehicle. After sleep deprivation, we found that wake SWA was, if anything, decreased under 5-MeO DMT as compared to vehicle. However, we have not previously analysed subsequent sleep after SD, and have done this now.

Remarkably, we found that the initial rebound of NREM SWA was reduced when preceded by SD+5-MeO DMT as compared to SD+vehicle. These results support the notion that 5-MeO-DMT slow waves may contribute to dissipation of homeostatic sleep pressure, potentially strengthening our conclusions of similarities between 5-MeO DMT-induced and spontaneous sleep slow waves. We now included an additional panel in Figure 3, and discuss these new results in the Results (page 5) and Discussion (page 6-7).

2) The discussion makes several claims about neural plasticity, particularly in connection to sleep-like mechanisms in association with 5-MeO-DMT. However, plasticity was not directly investigated in this study, and the purported link to sleep features remains speculative. These interpretations would benefit from a more cautious tone or explicit framing as hypotheses for future investigation. Similarly, the statement “we surmise that the 5-MeO-DMT induced slow waves have a distinct mechanism from bona fide sleep slow waves” seems not entirely justified by the findings, because 5-MeO-DMT-associated slow waves did show the off-periods as natural slow waves during NREM sleep.

We completely agree with the reviewer and revised the corresponding sentences accordingly.

3) Were other important features of sleep architecture and microarchitecture, such as microarousal density and dynamics, and NREM and REM sleep bout lengths assessed following the drug administration?

In our initial submission, we primarily focused on the immediate effects off 5-MeO DMT and detailed analysis of sleep architecture was omitted. We completely agree that it will strengthen our manuscript, even that the results were largely negative, and we now include further analysis of sleep bouts and brief awakenings, as well as mention briefly these results in the appropriate section (page 3).

4) The experiment using the 5-HT1A antagonist is highly relevant to understanding serotonergic mechanisms of sleep. Did the authors explore the sleep features after these manipulations? Additionally, did the authors analyze the behavioral changes of the animals in this condition?

We thank the reviewer for this question, but would like to clarify that in this experiment, we have decided to focus on acute effects of the drugs only in animals wearing the oculometer, as the animals had to be restrained to remove the devices at the end of the session which could have been stressful, and will have effects on subsequent sleep. The interaction of the drug effects with the potentially stressful manipulation cannot be ruled out, and therefore we prefer not to go in depth analysing these data as the conclusions may be misleading. In addition, we unfortunately do not have video recordings of sufficient quality for detailed quantitative analysis of their behaviour. We therefore can only comment on our visual observations, where we did not detect any obvious abnormalities, but unfortunately further quantitative analysis is not possible. We included a new supplementary video to illustrate that the behaviour of the animals was completely normal (Movie S5 and S6). We discuss this limitation in the revised version of our manuscript.

5) All power dynamics (e.g., Fig. 3c and similar) should be calculated relative to the vigilance state from which it is derived (Franken et al., 1991; Fernandez et al., 2018; Hubbard et al., 2020, Figure 1). For instance, to calculate homeostatic Delta decay during sleep, it should be done by dividing the recording into bins of equal amount of NREM; Then, the average delta power for each bin is computed, including only the NREM epochs. A second normalization to the values between ZT8-12, to the first hour, or similar is also commonly used (Fernandez et al., 2018 Sup Figure 4b). Otherwise, the dynamics might reflect or be contaminated by differences in sleep architecture. In fact, this might be the reason why the authors do not

observe the well described “SWA” decay after sleep deprivation or from the beginning of sleep.

We would like to reassure the reviewer that our approach of binning the data and power normalisation follows best standards in the field and our extensive experience with data analysis. While the approach to bin the data to allow equal amount of NREM sleep per interval could be appropriate in some cases, in our experiments it did not make a noticeable difference. However, we prefer to not use this approach based on our observation that the drug affects vigilance states and therefore comparing non-equivalent time intervals in relation to circadian phase or light-dark cycle becomes a possibility we wanted to avoid. Having said that, we always ensure that a sufficient amount of sleep is included in any of the bins analysed so that all individual animals are represented and contribute to the mean values and statistics. We added a paragraph in the Methods section to justify our approach for binning and normalising EEG power (page 15).

We also included the new analysis of NREM SWA time course after SD+5-MeO DMT and vehicle and found reduced values of SWA after the drug, raising an interesting possibility that 5-MeO DMT slow waves may contribute to dissipation of homeostatic sleep pressure (see our response above). We discuss these new findings in the Results (page 5) and Discussion (Page 6-7) of the revised manuscript.

6) As theta activity during wake has been related to exploration, and because the animals explored less after DMT, it raises the question of how much the reduction in theta is related to changes in the animal’s behavior. One alternative to check this is to correlate the changes in the theta to exploration or behavioral metrics during the post-injection period.

Overall, the animals retained relatively high level of locomotor activity and therefore we do not believe that the complete absence of theta-activity is simply a reflection of changes in behaviour. Our supplementary videos S1 illustrate that the animals were not immobile while experiencing 5-MeO DMT. However, the possibility cannot be excluded that the types of behaviour are very different and this explains changes in brain signals. We have already discussed this possibility in the submitted version of the manuscript and now elaborate on this further in the Results (page 3, 5) and the Discussion (page 10).

7) One particularly intriguing observation is the presence of fast-frequency activity superimposed on slow waves, as seen, for example, in Figure 1b. This pattern is reminiscent of EEG features observed in human sleep under the influence of certain drugs, and of some slow waves contained in micro-arousals in sleep. The authors are uniquely well-positioned to investigate and interpret these dynamics. Do these patterns reflect asynchronous activity across distinct neuronal populations—some exhibiting wake-like activity while others display sleep-like slow waves? Or might they originate from the same neuronal populations, perhaps reflecting increased firing during the upstate of the slow oscillation? What are the features of the fast activity? Is there any oscillatory (periodic) component associated with it? Which range of frequencies does it comprise? Is it only a reflection of the aperiodic components? The quantification and comparison of the MUA of NREM sleep slow waves and those related to the drug might also shed light on these matters in any case, an in-depth discussion and analysis of such ‘ultralocal’ aspects of sleep underlying the findings would be valuable here.

These are excellent questions, and we completely agree with the reviewer that our data are well suitable to pursue some of these aspects. We do not think however, that our approach of recording MUA, which was designed to simply obtain a general readout of cortical neuronal activity, is suitable for detailed analysis of how specific neuronal subtypes respond to the drug. We are indeed planning follow up studies to record and analyse MUA from different cortical layers and cortical regions, which will be ideal for answering the questions the reviewer is posing. We did perform additional analysis now and moved this figure from supplementary to Fig. 2 (page 4).

8) *What was the rationale for studying the overall SWA instead of the two components of this slow activity? Cortico-cortical Slow Oscillations (SO) and thalamo-cortical delta activity (Steriade 1994) have different dynamics across sleep, respond differently to sleep pressure (Hubbard et al., 2020), and thalamic manipulations (Fernandez et al., 2018; Osorio-Forero, et al., 2021). Furthermore, the local aspects of different slow waves might also differ as in humans (Siclari et al., 2014; Navarrete et al., 2023). Using at least two distinct bands would help strengthen the interpretation and claims of the study. Addressing the differences between SO and Delta is also essential for the interpretation of the fast-frequency activity superimposed on the slow waves and the experiments involving local 5-MeO-DMT infusion in the cortex.*

We completely agree with the reviewer that there is strong evidence for the two parts of the SWA band to have a different neurophysiological origin and dynamics. It is for this reason we always show the full spectra rather than or in addition to the time course of the entire SWA band. Our detailed analysis of LFP slow waves suggests that 5-MeO DMT slow waves have somewhat longer duration, i.e. lower frequency (this figure has now been modified to include averages rather than a representative animal only); however spectral analysis does not provide strong evidence for slower vs faster parts of the SWA range being drastically different. We discuss this further in the revised manuscript specifically highlighting that further studies are required to evaluate this aspect in greater detail (page 4, 7).

9) *When interpreting the regional findings (anterior vs. posterior findings), the authors may consider the regional distribution of serotonin. In the human cortex, serotonin metabolites follow a clear posterior-anterior gradient, highest in posterior cortical regions (Javoy Agid et al. 1989).*

We thank the reviewers for their comments and discuss this possibility in the revised manuscript (page 7-8).

Minor comments

10) *"State of arousal" in the introduction is an unusual term in this context. Consider using "behavioral state" or "vigilance state" for clarity.*

We checked and corrected or justified using specific terminology where appropriate.

11) *The term "intoxication" in the second paragraph of the introduction may imply overdose or pathology. Consider replacing it with "influence" or "administration" to avoid misleading connotations.*

Agreed and corrected

12) *Some information in the main text or the figures is missing, such as the sleep stage analyzed, for instance, in the slow waves analysis.*

We apologise for the omissions, which is now corrected.

13) *Concerning the non-oscillatory analysis, what is the rationale behind using this band? This analysis is not described in the methods, and differences in sleep stages are also not addressed across the groups.*

We thank the reviewer for pointing out this limitation, and acknowledge that we should have taken a better informed approach. Unfortunately, literature on the exact methodology to calculate spectral slope is rather inconsistent. In the previous version, we focused on the part

of the spectra close to the inflexion point at the intersection between wake and NREM sleep spectral curves, which we surmise may be significantly influenced by state-specific oscillations, such as spindle- and theta-activity. To address this limitation, we have now used a broader band between 20-120Hz. Interestingly, this had an effect on our overall conclusion. Previously we reported that for the frontal EEG, 5-MeO-DMT treatment was comparable to NREM sleep, but this finding was not confirmed by our new analysis, while we still find evidence for regional differences in the spectral slope. We have now revised the corresponding paragraphs in the manuscript (page 3) and elaborated on methodology in the Methods section (page 15).

14) For the cumulative time spent in REM, it is recommended to show the relative changes instead of the absolute cumulative time (McCarthy et al., 2016), as they might help to differentiate whether REM sleep homeostasis was altered or just shifted. This might be useful to address topics such as the one raised in: "One possibility is that the dissociated state induced by 5-MeO-DMT could replace some aspect(s) or functions(s) of REM sleep and thus reduce the homeostatic pressure to express REM sleep."

We thank the reviewer for this comment and now included a new panel showing relative changes in REM sleep, expressed in relation to the daily amount accumulated during baseline (Fig. S3.n).

15) For the MUA analysis, it is recommended to use a raster plot or a moving average, such as in Aston-Jones and Bloom, 2981 or Mann and Paulsen (2010).

All MUA are now provided as a raster plot and moved to Fig. 2.

16) In (Fig. 1.f-h-l, Movie S2), there is no l. (Fig. 1.j; Fig. S4.a, Movie S2), There is no "j".

All figures have been checked and corrected.

17) The movies are a highly valuable information. Merging Movies S1 and S2, showing the animal's behavior and concomitant signals, could be more informative. Otherwise, Movie S2 does not add further information to Figure 1a-b, and it would be more convincing to show that the slow waves appear simultaneously as the animal is moving. Also, the file of the last movie is broken or empty.

We corrected the file of the last movie. Regarding movie S1 and S2, we would like to point out that S1 shows an un-implanted animal and therefore it cannot be merged with S2, which shows recordings from implanted animal

18) The panels 1a-b and 1f-i might be redundant (with further examples in the supplementary data S7). For space purposes, including a complete description of the sleep stages with the LFP, EEG, EMG, and MUA activity might be sufficient.

We thank the reviewer for this recommendation, but these panels serve somewhat different purpose in our description of the effects, and we prefer to keep the figure as it is.

19) Concerning the methods

a. To support the observations, it is recommended that example images of the location of the cannulas and LFP electrodes be used, particularly for the experiment on local changes. Ideally for all animals, but at least a couple of examples.

We now include an image illustrating the position of the electrode and the cannula in the supplementary material.

b. What was the dosage of the oral Metacam?

We now mention the dose of Metacam in methods (page 12).

c. The “Epochs containing mixed frequencies of lower amplitude and high EMG between two sleep epochs (NREM–NREM or REM–NREM) were scored as movement” of < 12 - 16 seconds are usually studied as Microarousals (except for the REM–NREM transition), is there any reason why this nomenclature was not used, also, what was the rationale not to study changes in MAs in the study, particularly given that some discussion and conclusions were related to arousal.

This is a misunderstanding: we refer to those epochs as micro-arousals (or brief awakenings), and “movement” is simply how it is designated in our scoring program. We now clarify these criteria and terminology used to define these events in the Methods section (page 14).

d. Concerning the slow-wave detection, it would be crucial to show examples of the detected waves. Similarly, did using a standard-deviation-based threshold change the type of waves that could have been detected? Which kind of slow waves were detected during wake without DMT? Why were no additional time features, such as time between 0 crosses, used for the detected waves?

We include the average detected slow wave in Fig. 2, and raw data examples was provided in Figure S7. We have now included further illustration of the approach for slow wave detection, and further discussion of the properties of the detected slow waves (page 16). We used the similar approach as in our previous work (See Harding et al., 2023 BMC Neuroscience, McKillop et al., 2018 J. Neuroscience), where duration of the slow wave was not considered as a specific feature. In our opinion, plotting the distribution of relevant features of detected events, as we did on Figure 2 is more informative than imposing arbitrary thresholds to decide what events are bona fide slow waves and which are not.

e. No methodological description of the aperiodic analysis was provided.

We apologise for the omission, and this has now been corrected (page 15).

f. The authors did not show differences in behavior, sleep, etc., in animals with the oculometer compared to animals without the system.

As explained above, unfortunately we do not have video data of sufficient quality to provide robust quantitative analysis of the behaviour of animals wearing an oculometer. We discuss this now as a limitation, but would like to reassure the reviewer that no abnormalities were detected (page 10). We have added a new Supplementary video as a representative example.

20) In regard to the figures, the idea of providing a result description for every panel might be confusing unless defined by the journal; it is not encouraged. Concerning specific figures, a. Figure 1b: Please clarify whether the NREM and REM traces reflect natural sleep or sleep occurring under the influence of the drug. Also, specify the polarity convention (i.e., is negative up or down?). The last sentence, “The mean of all non-representative data is plotted with SEM,” only refers to d and e?

We follow the standards of the journal for figure legends, and now include further relevant information in the manuscript, specifically to answer the question on when the epochs from NREM and REM sleep were recorded.

b. Figure 2: Is the vehicle in b and c with or without SD?

This has now been clarified in the Figure legend, as it is without SD.

*c. Figure 4: The title “The effects of 5-MeO-DMT are mediated by 5-HT1A receptors.” is misleading. The results show that some effects are mediated by these receptors, but others are not. In this figure, it is also recommended to homogenize the panel dimensions. As mentioned before for Figure 3, it is essential to control for sleep stages when describing dynamic power changes. The result “The suppression of the occipital theta activity occurring after the injection of 5-MeO-DMT was largely suppressed by an injection of WAY resulting in vehicle-like levels of theta activity throughout interest (ME analysis, effect of time point*condition $F_{87,143} = 2.92$, $p < 0.0001$; Fisher’s LSD $p > 0.05$.” is not commented in the manuscript.*

We completely agree and the now Figure 5 title has been now modified accordingly and the results are further discussed in the manuscript.

d. Figure S1: The NREM trace appears unusually jagged—could this be an artifact? If not, some explanation would be helpful.

We agree this is unusual and this effect might have been caused by an improper filtering. The issue has been fixed as shown in Figure S1.

e. Figure S3: “Latency is expressed from the time the bowl is deposited in the home cage and the hand of the experimenter is removed from the environment” might better be described in the methods. What is the n for each experiment? (it is recommended to add it for each panel, or group of experiments) And why is there no difference in f if the values seem quite different? It is recommended to show each data point in the plots.

Figure legends have been amended to follow the editorial policy, and further discussion is provided in the revised manuscript. As the figure now show boxplots and individual values, we hope the non-significant results in S3f becomes clearer: we believe this is due to the low sample size and significant variability between individuals.

f. Figure S4: It is recommended to show the grand average for all animals in addition to the representative animal, a longer window around the peak of the waves, and representative examples of single waves in the three conditions (BL NREM, vehicle wake, and 5-MeO-DMT). What is the count, and not only the percentage of waves? Did they differ in number? They seem very different, in regarding the wake condition in the vehicle.

We thank for this comments, which has now been addressed and the new now Figure S2 has been completely reworked to include grand average rather than representative data. The count did differ, as NREM represents a larger proportion of recording time as opposed to vehicle and 5MeO-DMT conditions which, to remain comparable, include slow waves detected within the first hour following the injection. To account for this, we plot relative data as we have done in our earlier publications.

g. Figure S5: It is recommended to show the wake and the sleep-related FFT. Also for Figure 1.

We thank the reviewer for this recommendation, and the analysis for figure 1 has now been included in the revised manuscript (Figure S1). We have to point out that the experimental design of the experiment presented in figure S5 and S6 did not allow including undisturbed wake and sleep-related FFT.

h. Figure S7: There is no 2.2 in the figure. As mentioned, “e” might be redundant; you already have examples in Fig. 1. Additionally, add the FFT of all sleep stages for all manipulations instead.

We apologise for this omission, which is now corrected, and have made an effort to include EEG spectra for all experiments where relevant if the data are available.

References

- Steriade, M. (1994). Sleep oscillations and their blockage by activating systems. *Journal of Psychiatry and Neuroscience*, 19(5), 354.
- Hubbard, J., Gent, T. C., Hoekstra, M. M., Emmenegger, Y., Mongrain, V., Landolt, H. P., ... & Franken, P. (2020). Rapid fast-delta decay following prolonged wakefulness marks a phase of wake-inertia in NREM sleep. *Nature communications*, 11(1), 3130.
- Fernandez, L. M., Vantomme, G., Osorio-Forero, A., Cardis, R., Béard, E., & Lüthi, A. (2018). Thalamic reticular control of local sleep in mouse sensory cortex. *Elife*, 7, e39111.
- Navarrete, M., Osorio-Forero, A., Gómez, A., Henao, D., Segura-Quijano, F. E., Le Van Quyen, M., & Valderrama, M. (2023). Response of sleep slow oscillations to acoustic stimulation is evidenced by distinctive synchronization processes. *Sleep*, 46(6), zsad110.
- Siclari, F., Bernardi, G., Riedner, B. A., LaRocque, J. J., Benca, R. M., & Tononi, G. (2014). Two distinct synchronization processes in the transition to sleep: a high-density electroencephalographic study. *Sleep*, 37(10), 1621-1637.
- Franken, P., Dijk, D. J., Tobler, I., & Borbély, A. A. (1991). Sleep deprivation in rats: effects on EEG power spectra, vigilance states, and cortical temperature. *American Journal of Physiology-Regulatory, Integrative and Comparative Physiology*, 261(1), R198-R208.
- Javoy-Agid, F., Scatton, B., Ruberg, M., L'heureux, R., Cervera, P., Raisman, R., ... & Agid, Y. (1989). Distribution of monoaminergic, cholinergic, and GABAergic markers in the human cerebral cortex. *Neuroscience*, 29(2), 251-259.
- Osorio-Forero, A., Cardis, R., Vantomme, G., Guillaume-Gentil, A., Katsioudi, G., Devenoges, C., ... & Lüthi, A. (2021). Noradrenergic circuit control of non-REM sleep substates. *Current Biology*, 31(22), 5009-5023.
- McCarthy, A., Wafford, K., Shanks, E., Ligocki, M., Edgar, D. M., & Dijk, D. J. (2016). REM sleep homeostasis in the absence of REM sleep: effects of antidepressants. *Neuropharmacology*, 108, 415-425.
- Aston-Jones, G., & Bloom, F. (1981). Activity of norepinephrine-containing locus coeruleus neurons in behaving rats anticipates fluctuations in the sleep-waking cycle. *Journal of Neuroscience*, 1(8), 876-886.
- Mann, E. O., & Paulsen, O. (2010). Local field potential oscillations as a cortical soliloquy. *Neuron*, 67(1), 3-5.

Reviewer #3 (Remarks to the Author):

The manuscript reported that the injection of serotonin psychedelic, non-specific 5-HT receptor agonist 5-MeO-DMT induced a transient dissociated state of vigilance, characterized by coexistence of sleep-like slow oscillations in the brain signal and marked pupil dilation in freely behaving awake mice. The authors employed multiple approaches, including electrophysiological vigilance state recordings, LFP, pupil diameter monitoring using custom-built oculometer, power spectral slope analysis, behavioral video recording, intracortical injection. The study design is logical and well-structured and supported the previous reports using 5-MeO-DMT (Bréant, B. J. B. et al. Sleep Med. 100, S22 (2022).; Souza, A. C. et al. Sci. Rep. 14, 11281 (2024)).

We thank the reviewer for their positive evaluation of our manuscript.

1) While slow waves appear on the EEG following 5-MeO-DMT administration, they are

clearly different than one during slow wave sleep. Fig. 1i, the slow waves seem to have a higher frequency, and a lower amplitude as compared to SWS (Fig. 1g). It would be useful to have a direct comparison between SWS and post 5-MeO-DMT FFT analysis.

We agree with the reviewer that wake slow waves following 5-MeO administration show some similarities and differences from sleep slow waves. The comparison is not straightforward as slow waves within NREM sleep can be also extremely variable, and, for example, different between high and low sleep pressure. We had already provided a comparison of slow wave characteristics for a representative individual mouse, which has now included all animals (Fig. 2), and the comparison of FFT is now provided on several figures, specifically for both baseline condition and sleep deprivation (Fig. 3).

2) The use of a single 5 mg/kg dose of 5-MeO-DMT is justified based on prior studies reporting increased head twitch response at this concentration. However, incorporating multiple doses to assess dose-dependency would significantly strengthen the understanding of 5-MeO-DMT-generated slow waves.

We agree with the reviewer that further investigation of different doses could further strengthen our understanding of 5-MeO generated slow waves; however, we do not think it is directly relevant for the key aims and main conclusions of this paper. We are indeed planning a follow up study with additional doses, brain regions and cortical layers, but this will be a separate project.

3) "Sleep" should always follow REM and NREM in the text, figures and figure captions. If an abbreviation is needed, the abbreviation for REM sleep should be REMS and the abbreviation for NREM sleep should be NREMS. REM stands for rapid eye movement which is not a sleep stage.

We completely agree with this recommendations, and it has been amended throughout the manuscript.

4) Video 2: The EEG seems to not be detected during NREM sleep and REM sleep, only a line is seen. On the opposite, during wakefulness and post 5-MeO-DMT injection, the EEG fluctuations are clearly seen. REM sleep is labeled as PS and NREM sleep is labeled as SWS, which can be confusing for non-sleep experts. The manuscript is using REM sleep and NREM sleep, the video should keep these denominations.

We thank the reviewer for pointing this out, and it has been now corrected.

5) Regarding the Supplementary Videos demonstrating the coexistence of active behaviors and slow oscillations in brain signals, would it be possible to integrate the waking behavior along with the corresponding EEG and LFP signals from a single mouse into one synchronized video? This could provide a more intuitive understanding of the temporal relationship between behavior and neural activity.

We agree with the reviewer, but unfortunately we are unable to provide such a video. During the experiment, the top video recording setup was removed as it had a negative impact on the quality of the signals, introducing noise. Although the purpose of supplementary video S1 is to show the absence of abnormal behaviour, supplementary video S2 shows wake with 5-MeO-DMT, including slow waves and high muscle tone with some bursts in the EMG, indicating movement comparable to wake after vehicle.

6- Fig. S3f-g: Statistical analysis did not show a significant difference in average speed and maximum speed. However, the charts show a trend to decrease. The video S4 also indicates that the mice are spending less time running the wheel. According to the method section,

only 4 mice were included in the study. This number is probably too low. Interestingly, these parameters are also affected by the control injection, indicating that animal handling and the pain from the injection affect these behaviors. We can wonder if the protocol of the motor test is appropriate. In fact, the sample size (n=4 per group) is small for all behavioral paradigms assessing exploratory behavior, feeding, and locomotor activity. Increasing the number of animals would enhance the statistical power of the findings.

As discussed above, we would like to clarify that the primary purpose of this experiment was to rule out obviously abnormal behaviour such as reflecting serotonin syndrome. For this, a large number of animals was considered unnecessary, as with n=4 in this particular experiment and visual /video observations in other experimental series we were never able to detect gross abnormalities. We now clarify the aim of this experiment and acknowledge that the low number of animals precludes reliable statistical comparison (page 4).

7) To address the origin of the induction of SWA, the authors did not observe any marked behavioral or EEG alterations by intracortical infusion of 5-MeO-DMT in the primary somatosensory cortex and argue that the effects on brain state and arousal are more likely to be generated at the global level. Please clarify why the prefrontal cortex was not the target of choice given that slow waves are originating from the prefrontal cortex.

It is difficult to target the prefrontal cortex with surface screw electrodes and when we designed and started this experiment we did not expect to find the occurrence of slow waves in awake animals after injection of 5-MeO DMT. This result was completely unexpected, while the original aim of this study was to characterise the effects of the drug on sleep-wake states. We are very interested to continue this work, and planning to include other cortical regions in follow up studies, which will help us to provide a better understanding of the origin of slow waves induced by 5-MeO DMT.

8- Co-treatment with 1 mg/kg of the 5-HT1A receptor antagonist WAY did not counteract some of the effects of 5-MeO-DMT and instead further increased SWA. Please clarify the rationale for selecting the 1 mg/kg dosage of WAY. As WAY is not a selective 5-HT1A receptor antagonist, the authors should discuss the possible use of 5-HT2A-R and 5-HT1A-R knockout mice.

Previous studies show that WAY100635 is not a selective 5HT1A receptor antagonist, and we agree with the reviewer that the use of 5HT2A or 1AR KO mice would provide a better understanding of the mechanisms underlying the effects of 5MeO-DMT, and we will consider using this model in our follow up projects. We have now provided further justification for the choice of this pharmacological approach and the dose used (page 10, 12).

9) Some descriptions in the Results section appear to rely on qualitative observations rather than quantitative statistical analysis. Incorporating appropriate quantitative analyses would improve the scientific rigor of the findings. For example: Page 3, Plotting the EEG spectrograms following vehicle and 5-MeO-DMT revealed that marked effects of the compound on cortical activity lasted less than an hour in all cases (representative example: Fig. 1.c). Specifically, we observed that while the animals were awake and moving after the injection of 5-MeO-DMT, theta-frequency activity was replaced by slow frequencies for approximately 45 min before returning to levels comparable to vehicle (Fig. 1.c; Movie S2)

We agree with the reviewer and now provide further quantitative characterisation of the effects observed as appropriate (page 3).

10) Fig. S4b-c: The figure caption indicates that NREM sleep and wakefulness after the injection of vehicle or 5-MeO-DMT are presented but the chart legend indicates "Baseline NREM, Vehicle and 5-MeO-DMT".

We have corrected this now.

11) The analysis of power spectral slopes in the 20–30 Hz gamma band across conditions is intriguing. However, it would be helpful to clarify why only studies this specific frequency range.

Reviewer 2 has raised a similar point (please see above). We have now provided further justification of the frequency bands and performed the analysis on a broader frequency range which strengthen our results.

12) It will be useful to have the number of animals included in each experiment in the figure captions.

Done

13) The figure order can be optimized so they are cited in order in the text.

The figure order has been checked and optimised.

*14) Please include the full term for this abbreviation at first use:
5-HT 2A/2c agonist 1-(2,5-dimethoxy-4-iodophenyl)-2-aminopropane (DOI)*

Done

We thank the reviewers for their positive feedback, and we are delighted to hear that our manuscript is provisionally accepted in Communications Biology, pending the remaining minor revisions. Please find below our responses to the remaining comments.

Reviewer #2 (Remarks to the Author):

Why did you choose 20-120 Hz for the aperiodic activity? Additionally, do you fit in the linear-linear space, the Linear-Log Space, or the Log-Log space? The first one is uncommon and may be the reason why the values differ significantly from those in the literature.

We selected the 20–120 Hz range because it is the least affected by strong state-specific oscillatory activities prominent during waking and sleep in lower frequencies, such as sigma (10–15 Hz), theta (6–9 Hz), and delta (0.5–4 Hz) range, with the latter two directly affected by 5-MeO-DMT administration in our study. We agree that fitting the spectra in linear–log space is more conventional, and we have re-calculated the values accordingly to enable direct comparison with other studies. The changes are now reflected in the Methods, Results, and the legend for Supplementary Table 1.

Related to the experiment of local injection of DMT, in contrast to the author's conclusion, "These results suggest that the prominent effects on the brain state and arousal are more likely to be generated at the global level". Another possibility is that it is local but in a subcortical (or another cortical) area. The results only suggest that it's not in the local S1 vicinity. A good example of this is the effects of Noradrenaline on spindles (an effect mediated by the thalamic terminals of the LC, but not the cortical ones).

We thank the reviewer for this recommendation, which we have now incorporated in the discussion.

Reviewer #3 (Remarks to the Author):

The authors have addressed the main concerns from the first submission. The revised manuscript is improved with clearer explanations and stronger evidence supporting the main conclusions.

There are only minor typos:

Supplementary Video 4: SWS and PS are used instead of NREM sleep and REM sleep respectively. That could be confusing to some readers.

We have double-checked Supplementary Movie 4, and confirmed that we are now using correct terminology, as recommended.

Page 5, the sentence "Firstly, we replicated the effects of the injection of 5-MeO-DMT in animals wearing the oculometer (Fig. S6)," should reference Fig S5.

Done

Fig. S7 is a method figure and is not showing "a minor increase in EEG SWA was observed after the infusion (Fig. S7)." Page 6

Done

Fig. S8 is not cited in the manuscript

Done, in the methods section.